# Efficient Reinforcement Finetuning via Adaptive Curriculum Learning

**Taiwei Shi** *taiweish@usc.edu*
*University of Southern California*

**Yiyang Wu** *wendywu2@andrew.cmu.edu*
*Carnegie Mellon University*

**Linxin Song** *linxinso@usc.edu*
*University of Southern California*

**Tianyi Zhou** *tianyi.zhou@mbzuai.ac.ae*
*Mohamed Bin Zayed University of Artificial Intelligence*

**Jieyu Zhao** *jieyuz@usc.edu*
*University of Southern California*

**Reviewed on OpenReview:** *https://openreview.net/forum?id=UEhpyq41b9*

## Abstract

Reinforcement finetuning (RFT) has shown great potential for enhancing the mathematical reasoning capabilities of large language models (LLMs), but it is often sample- and compute-inefficient, requiring extensive training. In this work, we introduce ADARFT (*Adaptive Curriculum Reinforcement Finetuning*), a method that significantly improves the efficiency of RFT through adaptive curriculum learning. ADARFT dynamically adjusts the difficulty of training problems based on the model's recent reward signals, ensuring that the model consistently trains on tasks that are challenging but solvable. This adaptive sampling strategy accelerates learning by maintaining an optimal difficulty range, avoiding wasted computation on problems that are too easy or too hard. ADARFT requires only a lightweight extension to standard RFT algorithms like Proximal Policy Optimization (PPO), without modifying the reward function or model architecture. Experiments on competition-level math datasets demonstrate that ADARFT improves convergence efficiency and reasoning performance. Given problem-level difficulty annotations, ADARFT reduces RFT training time by up to $2\times$ across data distributions and model scales, offering a more scalable and effective RFT framework.

## 1 Introduction

Reinforcement Finetuning (RFT) has emerged as a powerful technique for aligning large language models (LLMs) with task-specific goals, particularly in domains such as mathematics and code generation where correctness is well defined (DeepSeek-AI et al., 2025; OpenAI et al., 2024b). By optimizing a policy model with reward signals that reflect task success, RFT enables more targeted learning than supervised finetuning (SFT) alone. However, despite its promise, RFT remains sample-inefficient and computationally expensive. Its training involves repeated rollout generation, reward computation, and policy updates—making it costly and difficult to scale (Ahmadian et al., 2024; Kazemnejad et al., 2024; Li et al., 2024; Hu, 2025; Cui et al., 2025). Recent efforts to address RFT inefficiency have focused on algorithmic simplification (e.g., RAFT (Dong et al., 2023), GRPO (DeepSeek-AI et al., 2025), ReMax (Li et al., 2024)), and data-centric strategies

(e.g., LIMO (Ye et al., 2025), LIMR (Li et al., 2025)). While these approaches improve sample or compute efficiency, they often introduce trade-offs: algorithmic simplifications may increase variance or limit stability, and static data filtering or scoring can be brittle, computationally heavy, or model-specific. Moreover, most methods' success relies on fixed datasets or training schedules, which can be suboptimal in non-uniform or imbalanced data regimes. More recently, early efforts have introduced curriculum-like ideas into RFT. Staged curricula divide training into a few manually-defined phases of increasing difficulty (Wen et al., 2025; Luo et al., 2025; Song et al., 2025), but these are coarse-grained and lack adaptivity. Other methods use online data filtering, repeatedly rolling out and pruning training samples until the model's average reward meets a target threshold (Bae et al., 2025; Yu et al., 2025). While this approach helps prevent the model from stagnating on problems that are either too easy or too difficult, it is not truly adaptive and incurs significant rollout overhead.

To address these limitations, we propose AdaRFT, a reinforcement finetuning method based on adaptive curriculum learning (Bengio et al., 2009), which dynamically adjusts training set difficulty to match the model's evolving skill level. The intuition is simple: learning is most effective when tasks are neither too easy nor too hard. AdaRFT formalizes this by maintaining a target difficulty level, which increases or decreases based on recent reward feedback. At each step, the model is trained on examples closest to this target, promoting a steady progression through solvable yet challenging tasks. The full algorithm is outlined in Algorithm 1. Unlike prior work that relies on fixed stages, repeated rollouts, or model-specific data processing, AdaRFT is lightweight, general, and model-agnostic. It can be directly applied on top of any standard reinforcement learning (RL) algorithms like Proximal Policy Optimization (PPO) (Schulman et al., 2017b). We evaluate AdaRFT on a dataset spanning a wide range of competition-level math problems, including AMC, AIME, and IMO-style questions. Across multiple training distributions and two model sizes, AdaRFT improves training efficiency while maintaining or often improving final performance in our runs. Gains are especially notable in imbalanced data regimes, where static sampling often fails. Given problem-level difficulty annotations, AdaRFT can reduce RFT training time by up to $2\times$, offering a practical path to more efficient RFT in structured reasoning tasks.

## 2 Related Work

**Efficient Reinforcement Finetuning.** Most RFT pipelines build on Proximal Policy Optimization (PPO) (Schulman et al., 2017b), with recent variants like RAFT (Dong et al., 2023), ReMax (Li et al., 2024), GRPO (DeepSeek-AI et al., 2025), and REINFORCE++ (Hu, 2025), aiming to reduce computational overhead by simplifying RL components. While effective, these methods often trade off stability or sample efficiency. In parallel, data-centric strategies have emerged as promising alternatives for efficient finetuning. LIMO (Ye et al., 2025) and s1 (Muennighoff et al., 2025) show that small, carefully selected supervised datasets can yield strong downstream performance, but their success hinges on manual curation, prompt engineering, and careful dataset construction, which may not generalize across tasks or models. LIMR (Li et al., 2025) and Wang et al. (2025) proposes scoring training examples based on their estimated learning impact, enabling selective finetuning with fewer samples. Yet, computing these scores requires a full training run, and the scores must be recomputed for each new model, limiting practicality and scalability. Moreover, reducing the number of training samples does not inherently translate to improved efficiency. Models still require a comparable number of optimization steps and wall-clock time to converge. In contrast, AdaRFT introduces a lightweight, model-agnostic curriculum learning strategy that dynamically adjusts task difficulty based on reward feedback. This allows continuous adaptation to the model's capabilities, improving convergence speed and final accuracy without modifying the RL algorithm or requiring manual data curation.

**Curriculum Learning for RL.** Curriculum learning (CL) structures training by presenting tasks in an organized progression, typically from easy to hard, to enhance learning efficiency and generalization (Bengio et al., 2009). In RL, CL methods include task sorting by difficulty (Zaremba & Sutskever, 2015; Justesen et al., 2018; Wang et al., 2019), teacher-student frameworks that adaptively select tasks based on learning progress (Matiisen et al., 2017; Portelas et al., 2019), and self-play approaches that induce automatic curricula through agent competition (Sukhbaatar et al., 2018; Zhao et al., 2025). Other strategies use intermediate-goal generation in sparse-reward settings (Florensa et al., 2018), unsupervised skill discovery (Jabri et al., 2019),

---

**Algorithm 1** ADARFT– Adaptive Curriculum Reinforcement Finetuning

---
1: **Input:** Data source $D$ with difficulty scores $\{d_i\}$, policy model $\pi_\theta$, reward function $R(\cdot, \cdot)$, batch size $B$, initial target difficulty $T$, step size $\eta$, sensitivity $\alpha$, target reward $\beta$, difficulty bounds $d_{\min}, d_{\max}$
2: Select RL algorithm $\mathcal{A}$ (e.g., PPO, GRPO, REINFORCE++)
3: **while** training is not finished **do**
4: $\quad \Delta_i \leftarrow |d_i - T| \quad \forall i \in \{1, \dots, |D|\}$ $\qquad\qquad$ ▷ Compute differences between problem and target difficulty
5: $\quad X \leftarrow \{s_1, s_2, \dots, s_B\}$ $\qquad\qquad\qquad$ ▷ Sort and select top $B$ samples closest to target difficulty
6: $\quad G \leftarrow \pi_\theta(X)$ $\qquad\qquad\qquad\qquad\qquad$ ▷ Generate responses using policy model
7: $\quad R_{\text{avg}} \leftarrow \frac{1}{|X|} \sum_{i=1}^{|X|} R(X_i, G_i)$ $\qquad\qquad\qquad\qquad$ ▷ Compute average reward
8: $\quad \pi_\theta \leftarrow \mathcal{A}(\pi_\theta, X, G, R)$ $\qquad\qquad\qquad\qquad\qquad$ ▷ Update policy
9: $\quad T' \leftarrow \text{clip}(T + \eta \cdot \tanh(\alpha \cdot (R_{\text{avg}} - \beta)), d_{\min}, d_{\max})$ $\qquad$ ▷ Update and clip target difficulty
10: $\quad T \leftarrow T'$ $\qquad\qquad\qquad\qquad\qquad\qquad\qquad\qquad$ ▷ Update sampler
11: **end while**

---

or knowledge transfer via progressive networks and imitation (Czarnecki et al., 2018; Rusu et al., 2022). While CL is well-studied in classical RL, its application to RFT of LLMs is still limited. Existing methods typically use staged training with hand-designed difficulty tiers (Wen et al., 2025; Luo et al., 2025; Song et al., 2025), or online filtering schemes that repeatedly sample and discard data until rewards reach a target range (Bae et al., 2025; Yu et al., 2025). These methods either lack adaptability or introduce significant computational overhead due to repeated rollouts. In contrast, ADARFT is among the first truly adaptive curriculum learning approaches for RFT: it continuously adjusts task difficulty based on the model's reward signal, enabling efficient, scalable training without fixed schedules or repeated rollouts.

## 3   AdaRFT

We aim to improve the performance of a policy model $\pi_\theta$ for solving mathematical problems through adaptive curriculum learning. Fine-tuning on problems that are too easy or too hard leads to poor learning outcomes. Instead, the model should be trained on problems whose difficulty is close to the model's current capability. We frame this as an adaptive curriculum learning problem and propose ADARFT, which adaptively adjusts the target difficulty to keep training problems within a suitable difficulty range. ADARFT is compatible with a variety of RL algorithms (e.g, GRPO, PPO); in this work, we instantiate it with PPO and refer to this variant as ADARFT (PPO).

Let $D$ be a dataset of mathematical problems, each annotated with a precomputed difficulty score $d_i$. The score can be either human-annotated or model-estimated. The objective is to train a policy $\pi_\theta$ that improves its problem-solving ability by dynamically adjusting the training curriculum according to the model's current performance. Our proposed algorithm, ADARFT, is shown in Algorithm 1.

### 3.1   Dynamic Curriculum Sampling

To construct an adaptive curriculum, we define a target difficulty $T$, which represents the current target difficulty level for training (more in Section 3.3). ADARFT dynamically adjusts $T$ based on the model's reward signal to maintain an optimal difficulty level for learning. At each step, the algorithm computes the absolute difference between the target difficulty and the difficulty of each problem in the dataset (Alg. 1, line 4): $\Delta_i = |d_i - T|$ for all $i \in [1, |D|]$. The batch of training problems is formed by selecting the $B$ problems with the smallest values of $\Delta_i$ (Alg. 1, line 5), producing a batch: $X = \{s_1, s_2, \dots, s_B\}$. This deterministic nearest-difficulty selection is intentional: when the model cannot solve problems around the current target difficulty after many retry attempts, sampling substantially easier or harder problems provides a weaker curriculum signal for the target capability region. This ensures that the selected problems are closest to the model's current target difficulty, focusing the learning process on problems that are neither too easy nor too hard. A natural extension is to sample from a Gaussian or other smooth distribution centered at $T$, which may improve diversity while retaining the target-difficulty bias; we leave this exploration to future work.

## 3.2 Policy Update

The selected batch $X$ is used to train the policy model $\pi_\theta$, which generates responses: $G = \pi_\theta(X)$. A reward signal is computed based on the correctness of the model's output (Alg. 1, line 7): $R_i = 1$ if the response is correct, and $R_i = 0$ if the response is incorrect. The average reward over the batch is computed as (Alg. 1, line 7): $R_{avg} = \frac{1}{|X|} \sum_{i=1}^{|X|} R(X_i, G_i)$. The policy can then be updated using a reinforcement learning algorithm $\mathcal{A}$ such as PPO, GRPO, or REINFORCE++ (Alg. 1, line 8): $\pi_\theta \leftarrow \mathcal{A}(\pi_\theta, X, G, R)$.

## 3.3 Target Difficulty Update

To adapt the curriculum dynamically, the target difficulty is updated based on the average reward. If the model performs well on the current difficulty level (high reward), the target difficulty increases, making the training problems harder. Conversely, if the model performs poorly, the target difficulty decreases. This dynamic update mechanism lies at the core of ADARFT's curriculum adaptation strategy. The update rule (Alg. 1, line 9) is defined as:

$$T' = \text{clip}(T + \eta \cdot \tanh(\alpha \cdot (R_{avg} - \beta)), d_{\min}, d_{\max}).$$

The full derivation is provided in Appendix B.2. This update rule is a smooth and stabilized variant of a standard linear mapping between reward space and difficulty space. Here, $\eta, \alpha, \beta$ are hyperparameters: $\eta$ is the step size for adjusting the target difficulty, $\alpha$ controls the sensitivity of the update, and $\beta$ is the target reward level, representing the desired success rate. The tanh function ensures smooth updates and prevents large jumps in difficulty by saturating for large deviations, while the "clip" function constrains the target difficulty within the valid range $[d_{\min}, d_{\max}]$. These bounds can be manually specified or automatically derived from the training set, for example, by taking the minimum and maximum of the difficulty scores $\{d_i\}$. Intuition and guidance for selecting these hyperparameters are discussed in Section 3.4, 4.3, and Appendix B.2.

## 3.4 Motivation for Target Reward $\beta$

A key component of ADARFT is its adaptive curriculum mechanism, which steers training toward a target reward level $\beta$. Intuitively, we aim to train on examples that are neither trivially easy nor prohibitively hard. In this light, setting $\beta = 0.5$, corresponding to a success rate of roughly 50%, naturally aligns with this goal. This section formalizes that intuition by analyzing the relationship between reward variance and learnability in RFT with binary rewards.

In entropy-regularized reinforcement learning, the optimal policy $\pi^*$ can be expressed relative to a reference policy $\pi_{\text{init}}$ as (Korbak et al., 2022; Go et al., 2023; Rafailov et al., 2023):

$$\pi^*(y \mid x) = \frac{1}{Z(x)} \pi_{\text{init}}(y \mid x) \exp\left(\frac{1}{\tau} r(x, y)\right) \tag{1}$$

where $\tau$ is the inverse temperature parameter controlling entropy regularization, and $Z(x)$ is the partition function that normalizes the action probability. The corresponding optimal value function and the partition function is given by (Schulman et al., 2017a; Richemond et al., 2024):

$$V^*(x) := \tau \log \mathbb{E}_{y \sim \pi_{\text{init}}(\cdot \mid x)} \left[\exp\left(\frac{1}{\tau} r(x, y)\right)\right], \quad Z(x) = \exp\left(\frac{1}{\tau} V^*(x)\right). \tag{2}$$

We can then take the expectation of the log-ratio between the optimal policy and the initial policy with respect to $y \sim \pi_{\text{init}}(\cdot \mid x)$, leading to (Haarnoja et al., 2017; Schulman et al., 2017a):

$$\mathbb{E}_{y \sim \pi_{\text{init}}(\cdot \mid x)} \left[\log \frac{\pi^*(y \mid x)}{\pi_{\text{init}}(y \mid x)}\right] = \frac{1}{\tau} \mathbb{E}_{\pi_{\text{init}}}[r(x, y)] - \frac{1}{\tau} V^*(x). \tag{3}$$

Since the left-hand side can be interpreted as the negative reverse KL divergence between $\pi_{\text{init}}$ and $\pi^*$ (Rafailov et al., 2024), Bae et al. (2025) show that when the reward $r(x, y)$ with $y \sim \pi_{\text{init}}(\cdot \mid x)$ is Bernoulli,

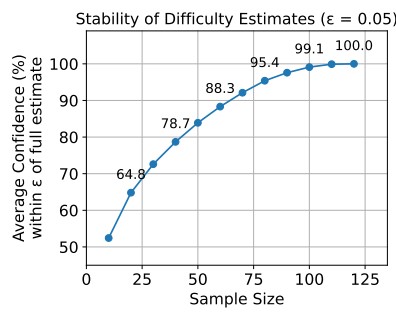
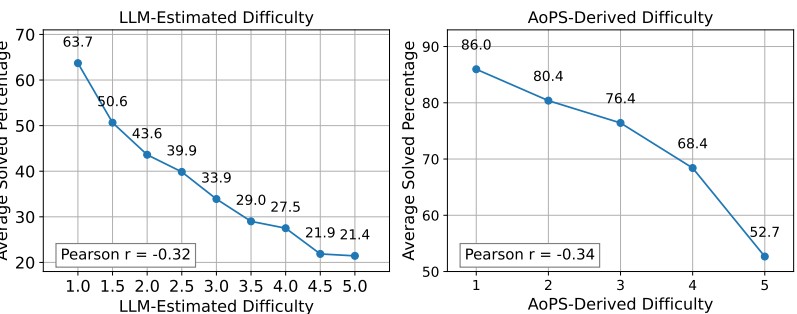

(a) Average confidence that subsampled difficulty estimates fall within ±0.05 of the full-sample estimate.

(b) Correlation between average solved percentage and two types of difficulty labels: (left) LLM-estimated difficulty and (right) AoPS-derived difficulty levels.

Figure 1: Evaluation of difficulty estimation: (a) Stability of difficulty scores under subsampling of model rollouts; (b) Correlation between labeled difficulty levels and average solved percentage.

the KL divergence is lower-bounded by the reward variance:

$$D_{\mathrm{KL}}(\pi_{\mathrm{init}}\|\pi^*) \geq \frac{p(x)(1-p(x))}{2\tau^2} \tag{4}$$

where $p(x)$ is the model's success rate on prompt $x$. This implies that the lower bound on the KL divergence is proportional to the reward variance, which is maximized when $p(x) = 0.5$. We use this result as an intuition for why prompts that the model succeeds on roughly half the time can provide informative policy-gradient updates; it does not by itself prove that PPO gradient norm or downstream learning progress is globally maximized at $p(x) = 0.5$. In Section 5 and Appendix 5.3, we conduct an ablation study by varying the target reward $\beta$, demonstrating that setting $\beta = 0.5$ consistently leads to the best performance, empirically supporting the use of an intermediate-success curriculum.

## 4 Experiments

### 4.1 Difficulty Estimation

Accurate estimation of problem difficulty is critical for ADARFT. For difficulty estimation, we select the Qwen 2.5 MATH 7B model (Qwen et al., 2025) because it demonstrates a balanced solving ability. A model that is too strong (e.g., OpenAI o1 (OpenAI et al., 2024b), DeepSeek R1 (DeepSeek-AI et al., 2025)) would solve most problems on the first attempt, leading to poor discrimination between easy and hard problems. Conversely, a model that is too weak (e.g., LLaMA 3.3 1B (Grattafiori et al., 2024)) would fail to solve most problems even after multiple attempts, limiting the signal required for curriculum adaptation. For each problem, the difficulty score is computed as: $d_i = 100(1 - s_i/n)$, where $s_i$ denotes the number of successful attempts on problem $i$, and $n$ is the number of attempts per problem. In our setup, we use $n = 128$.

To evaluate the stability of our difficulty estimation process, we simulate how confidence varies with different numbers of samples. For each problem, we treat the full set of 128 rollouts as the ground-truth difficulty estimate and compute how often sub-sampled estimates fall within a tolerance of $\epsilon = 0.05$. Specifically, we run 10 random sampling trials per sample size and average the confidence across all problems in the dataset. As shown in Figure 1a, even with as few as 64 samples, the estimated difficulty remains within ±0.05 of the full estimate over 90% of the time. With just 40 samples, the confidence remains around 80%. These results indicate that accurate and robust difficulty estimation can be achieved with fewer rollouts, reducing the computational burden of large-scale curriculum construction.

To further validate the reliability of our difficulty estimates, we examined their alignment with the difficulty levels provided in the MATH dataset. The MATH dataset comprises 12,500 competition-level mathematics problems sourced from contests such as the American Mathematics Competitions (AMC) and the American

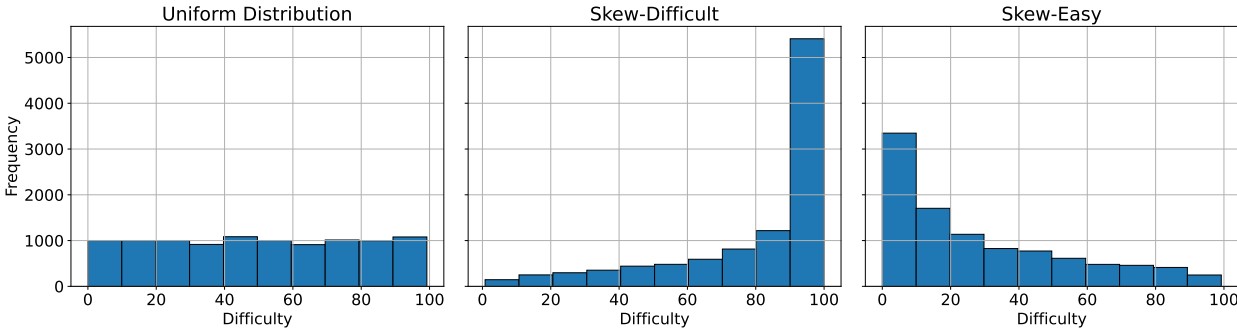

Figure 2: Difficulty distribution for different training sets: Uniform, Skew-Difficult, and Skew-Easy. Each training set contains 10,000 samples.

Invitational Mathematics Examination (AIME). Each problem is categorized into one of five difficulty levels, following the classification system used by the Art of Problem Solving (AoPS) community (details in Appendix B.3). In this system, level 1 denotes the easiest problems, while level 5 represents the most difficult. As shown in Figure 1b, there is a clear downward trend in the average solve rate as the labeled difficulty level increases, ranging from 86.0% at level 1 to 52.7% at level 5. Specifically, the AoPS-derived difficulty levels yield a Pearson correlation of $r = -0.34$ ($p < 0.05$) with model success rates. This negative correlation indicates that the model's empirical performance aligns well with the intended difficulty stratification, reinforcing the utility of both the labeled difficulty levels and our estimation approach in guiding curriculum learning. To further streamline the difficulty estimation process, we also prompted GPT-4o (`gpt-4o-0806`) (OpenAI et al., 2024a) to assign difficulty levels to the DeepScaleR dataset based on the AoPS rubric. Each problem was presented to GPT-4o with a request to rate its difficulty according to AoPS guidelines (the full prompt is shown in Appendix B.3). This approach provides a lightweight and scalable alternative to rollout-based estimation. As shown in Figure 1b, GPT-4o's difficulty ratings also correlate well with the model success rates, with a Pearson correlation of $r = -0.32$ ($p < 0.05$), making it a practical proxy for curriculum scheduling when computational resources are constrained.

## 4.2   Dataset

We use the DeepScaleR dataset (Luo et al., 2025) as the training set. DeepScaleR compiles problems from multiple sources, including AIME from 1984 to 2023 and AMC prior to 2023. The dataset also includes problems from the Omni-MATH (Gao et al., 2024) and Still datasets (Team, 2025), which feature problems from various national and international math competitions. This results in a diverse and challenging training set, covering a wide range of mathematical domains and difficulty levels.

In practice, we do not have control over the exact difficulty distribution of the data collected for training. This motivates our investigation into how different difficulty distributions influence ADARFT. To this end, we construct three distinct distributions from the DeepScaleR dataset. The first is a `skew-difficult` distribution, where most problems are challenging. The second is a `skew-easy` distribution, where most problems are relatively easy. The third is a `uniform` distribution, where problems are evenly balanced across all difficulty levels, ensuring a consistent representation of easy, moderate, and hard problems. Each of these three distributions includes 10,000 samples. The data distribution for each setting is shown in Figure 2.

For evaluation, we use six benchmark datasets to assess the model's performance across different levels of difficulty and mathematical reasoning. The first benchmark, MATH 500 (Lightman et al., 2023), is a subset of the MATH dataset (Hendrycks et al., 2021) containing 500 representative problems designed to test a model's general mathematical capability. GSM8K (Cobbe et al., 2021) is a set of grade-school math problems. OlympiadBench (He et al., 2024) includes a collection of problems from Olympiad-level mathematics and physics competitions. Minerva Math (Lewkowycz et al., 2022) is a curated set of undergraduate-level math problems that assess complex mathematical reasoning and symbolic manipulation. AMC 23 and AIME 24 include problems from the 2023 American Mathematics Competitions and the 2024 American Invitational

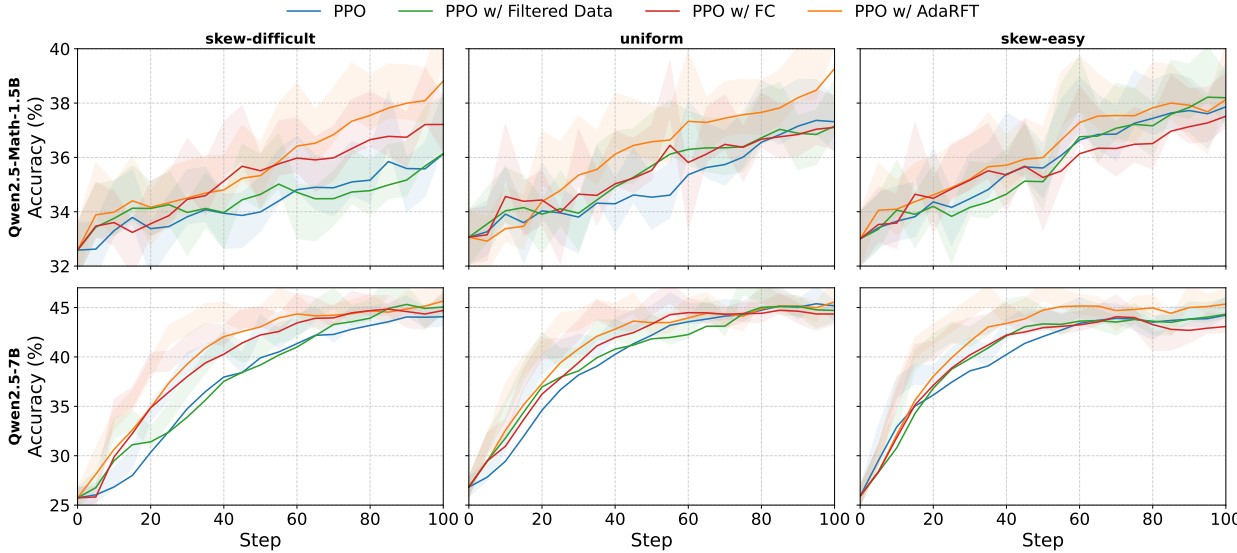

Figure 3: Performance comparison of PPO, PPO with filtered data, PPO with fixed curriculum (PPO w/ FC), and ADARFT (PPO) across different setups (uniform, skew-easy, skew-difficult). Accuracy is the average of MATH 500, GSM8K, AIME 24, AMC 23, OlympiadBench, and Minerva Math. Compared with baselines, ADARFT improves both the accuracy and training efficiency. For clarity, curves are exponentially smoothed.

Mathematics Examination, respectively. Since AMC 23 contains only 40 problems and AIME 24 only 30, we report accuracy as the average over 8 sampled responses per problem to ensure stable estimates. Together, these datasets span elementary, high school, and advanced competition-level math, providing a comprehensive evaluation of the model's reasoning abilities.

### 4.3 Training Setup

We trained two models on the three difficulty-based distributions of the DeepScaleR dataset described in Section 4.2: Qwen 2.5 7B and Qwen 2.5 MATH 1.5B. This setup allows us to evaluate the effectiveness of ADARFT on models with different initial performance levels when exposed to skew-difficult, skew-easy, and uniform problem distributions. All models were trained using four different approaches: (1) the standard PPO algorithm, (2) ADARFT (PPO), our method that integrates adaptive curriculum learning with PPO (see Section 3), (3) PPO with filtered data, a baseline that trains PPO on data filtered by pass@$k$ accuracy, and (4) PPO with a fixed curriculum schedule.

For the data filtering baseline (3), following prior work (Bae et al., 2025; Hu et al., 2025; Zyphra, 2025), we first run a pass@40 analysis for each combination of model and data distribution. We then discard examples that are either too easy or too hard, removing all problems with solved rates $\leq 10\%$ or $\geq 90\%$. This restricts training to problems of intermediate difficulty. However, this procedure removes a large fraction of the data, including many potentially informative examples. In addition, because difficulty is defined using pass@$k$ metrics, the filtering must be recomputed whenever the model or the data distribution changes.

For the fixed curriculum baseline (4), we follow the approach of prior work (Parashar et al., 2025; Kimi et al., 2025). In this setting, the difficulty of sampled problems follows a predetermined schedule that increases linearly over training steps. Suppose training runs for $m$ total steps. We then define a target difficulty $T(s)$ at step $s$ by $T(s) = T_{\min} + \frac{T_{\max} - T_{\min}}{m} \cdot s$, and at each step we sample problems whose estimated difficulty matches this target. Unlike ADARFT, this schedule increases difficulty at a fixed rate regardless of how quickly or slowly the model learns.

The training batch size was set to $B = 1024$, with the target reward $\beta$ set to 0.5 to promote learning at a balanced success rate. The sensitivity parameter $\alpha$ and step size $\eta$ were tuned using a validation set to ensure

| Model | Setup | Method | Final Acc (%) ↑ | Avg Step Time (s) ↓ | Extra Steps (%) ↓ | Extra Steps ↓ | Extra Time (s) ↓ |
|---|---|---|---|---|---|---|---|
| Qwen2.5 Math 1.5B | skew-difficult | ADARFT | **40.48** | **122.24** | **0.0%** | **+0** | **0.00** |
| | | PPO | 37.41 | 132.95 | 71.7% | +43 | 5716.85 |
| | | PPO (w/ Filter) | 37.22 | 128.20 | 81.7% | +49 | 6281.80 |
| | | PPO (w/ FC) | 37.22 | 130.91 | 26.7% | +16 | 2094.56 |
| | uniform | ADARFT | **41.11** | **121.31** | **0.0%** | **+0** | **0.00** |
| | | PPO | 37.20 | 126.82 | 56.7% | +34 | 4311.88 |
| | | PPO (w/ Filter) | 37.87 | 126.35 | 86.7% | +52 | 6570.20 |
| | | PPO (w/ FC) | 37.27 | 126.40 | 80.0% | +48 | 6067.20 |
| | skew-easy | ADARFT | **39.18** | 120.52 | **0.0%** | **+0** | **0.00** |
| | | PPO | 38.46 | 121.15 | 26.7% | +16 | 1938.40 |
| | | PPO (w/ Filter) | 38.15 | **115.12** | 35.0% | +21 | 2417.52 |
| | | PPO (w/ FC) | 38.09 | 121.99 | 58.3% | +35 | 4269.65 |
| Qwen2.5 7B | skew-difficult | ADARFT | **46.83** | **239.92** | **0.0%** | **+0** | **0.00** |
| | | PPO | 44.17 | 246.21 | 60.0% | +24 | 5909.04 |
| | | PPO (w/ Filter) | 45.35 | 254.22 | 62.5% | +25 | 6355.50 |
| | | PPO (w/ FC) | 45.54 | 243.12 | 22.5% | +9 | 2188.08 |
| | uniform | ADARFT | **46.92** | **234.16** | **0.0%** | **+0** | **0.00** |
| | | PPO | 44.70 | 243.82 | 32.5% | +13 | 3169.66 |
| | | PPO (w/ Filter) | 44.57 | 263.11 | 57.5% | +23 | 6051.53 |
| | | PPO (w/ FC) | 44.32 | 240.62 | 17.5% | +7 | 1684.34 |
| | skew-easy | ADARFT | **45.94** | 247.44 | **0.0%** | **+0** | **0.00** |
| | | PPO | 45.07 | 235.27 | 50.0% | +20 | 4705.40 |
| | | PPO (w/ Filter) | 44.98 | **233.13** | 42.5% | +17 | 3963.21 |
| | | PPO (w/ FC) | 43.46 | 240.66 | 57.5% | +23 | 5535.18 |

Table 1: Average training time per step (in seconds), final average accuracy (%) at step 100 averaged over MATH 500, GSM8K, AIME 24, AMC 23, OlympiadBench, and Minerva Math, and the additional training steps required for each baseline to match ADARFT 's accuracy at step 60 for Qwen 2.5 Math 1.5B or step 40 for Qwen 2.5 7B, across different setups and methods.

stable curriculum updates. We set $\alpha = 2$, $\eta = 50$, and the initial target difficulty $T = 0$. The step size $\eta$ acts as a scaling factor between the reward signal and the difficulty metric. Since the difficulty metric ranges from 0 to 100 and the reward ranges from 0 to 1, a target reward $\beta = 0.5$ implies that the maximum reasonable adjustment to the difficulty metric should be around 50. Therefore, we set $\eta = 50$ to scale the reward signal appropriately to the difficulty range. The sensitivity parameter $\alpha = 2$ controls the slope of the tanh function. Setting $\alpha$ to 2 makes the tanh function behave approximately linearly when the difference between the average reward and the target reward is small. The tanh function smooths out the adjustments, allowing for more controlled changes when the difference is large while maintaining sensitivity when the difference is small. Both models were trained on 8 A100 GPUs for approximately 100 steps. The implementation details can be found in Appendix B.

## 5 Results and Analysis

We evaluate the performance of standard PPO and ADARFT (PPO) across multiple training setups and two model sizes: Qwen 2.5 MATH 1.5B and Qwen 2.5 7B. Figure 3 presents the learning curves averaged across six benchmarks, while Table 1 and 2 provide a detailed breakdown of accuracy and training efficiency. On average, models trained with ADARFT (PPO) reach comparable accuracy in fewer RFT training steps and also obtain higher final accuracy in our single-seed runs. This improvement is particularly notable in non-uniform data distributions, where curriculum adaptation is most beneficial.

### 5.1 Training Efficiency

As shown in Figure 3 and Table 1, models trained with ADARFT consistently require fewer training steps to match the performance of those trained with standard PPO, PPO on filtered data, and PPO with a fixed curriculum schedule. Specifically, we report how many additional steps are needed for PPO variants to match the performance of ADARFT at step 60 for Qwen 2.5 Math 1.5B, and step 40 for Qwen 2.5 7B. Because models are evaluated only every 5 training steps, we apply exponential smoothing with a smoothing

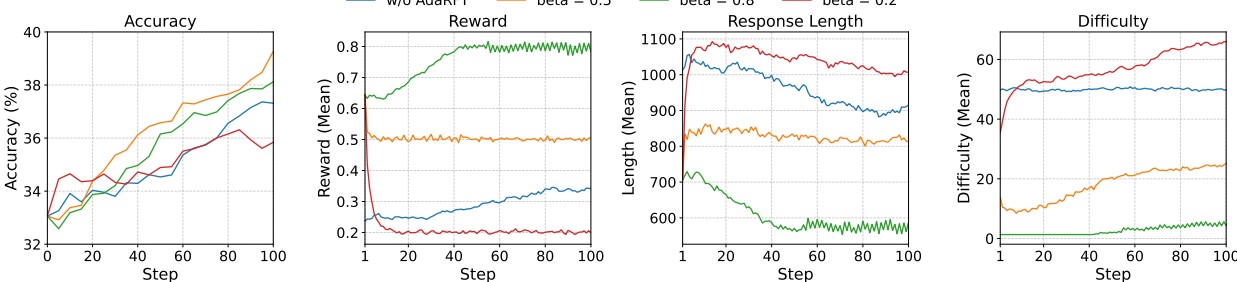

Figure 4: Ablation on $\beta$ in ADARFT: we compare model accuracy, average reward, response length, and mean difficulty under $\beta = 0.2$, $\beta = 0.5$, and $\beta = 0.8$, along with standard PPO (w/o ADARFT).

parameter of 0.3 to the accuracy curves to reduce variance. The shaded areas in Figure 3 represent the raw, unsmoothed accuracy $\pm 1\%$. For Qwen 2.5 Math 1.5B, standard PPO requires 43 extra steps (+71.7%) in the skew-difficult setting and 34 steps (+56.7%) in the uniform setting to match ADARFT's performance. PPO with filtered training data requires even more: +49 steps (81.7%) and +52 steps (86.7%) in the respective settings. In the skew-easy scenario, PPO requires +16 steps (26.7%), while PPO with filtered data needs +21 steps (35.0%) to catch up to ADARFT. The efficiency gains remain substantial with the larger Qwen 2.5 7B model. In the skew-difficult setting, PPO and PPO with filtered data require +24 steps (60.0%) and +25 steps (62.5%), respectively. PPO with a fixed curriculum schedule also follows this trend, suggesting that while fixed curricula can modestly improve training efficiency, their inability to adapt the the model's evolving learning dynamics limits their convergence speed relative to ADARFT.

In addition to improved sample efficiency, ADARFT also achieves faster average training time per step across nearly all settings, as reported in Table 1. This is largely due to the fact that easier problems require fewer tokens to solve. For example, an arithmetic reasoning question from GSM8K might require only around 200 tokens to reach a correct answer, whereas a competition-level math problem from AIME could require around 2000 tokens, a 10× difference in rollout length. The total token length affects multiple components of the training step, including the rollout itself and the subsequent PPO update. While PPO update time does not scale linearly with sequence length due to batching and attention computation patterns, longer sequences still incur higher compute costs. As a result, curriculum learning's tendency to prioritize easier problems early in training leads to shorter sequences on average, reducing per-step compute and improving overall training throughput.

## 5.2 Model Performance

While the primary goal of ADARFT is to improve training efficiency, it also consistently maintains and often improves final model performance in our runs. As shown in Tables 1, 2, and Figure 3, ADARFT achieves higher final accuracy at the end of training (step 100) across all configurations. Reported results are averaged over six benchmarks: GSM8K, MATH 500, OlympiadBench, Minerva Math, AMC 23, and AIME 24. On skew-difficult data, Qwen 2.5 Math 1.5B improves from 37.41% with PPO to 40.48% with ADARFT, a gain of over 3 percentage points. Similar gains are observed in the uniform setting (41.11% vs. 37.20%). Even on skew-easy data, where the baseline already performs well, ADARFT still yields modest improvements (39.18% vs. 38.46%). For the larger Qwen 2.5 7B model, improvements are smaller but consistent: from 44.17% to 46.83% on skew-difficult data, from 44.70% to 46.92% in the uniform setting, and from 45.07% to 45.94% on skew-easy data. Since these final accuracy results are based on one seed per configuration, they should be interpreted as empirical trends rather than statistically significant accuracy improvements. They support the conclusion that ADARFT improves training efficiency without sacrificing final accuracy. A more detailed per-benchmark breakdown is provided in Appendix A.2.

### 5.3 Ablation on Target Reward $\beta$

To better understand the role of the target reward $\beta$ in ADARFT, we perform an ablation study varying $\beta$ in the target difficulty update rule. Recall that $\beta$ controls the target average reward the model is expected to achieve and implicitly steers the curriculum: lower values prioritize easier problems, while higher values shift the curriculum toward more challenging samples. We train a Qwen 2.5 Math 1.5B model on the uniform data distribution with ADARFT (PPO) using three different values of $\beta$: 0.2, 0.5, and 0.8. For comparison, we also include standard PPO without ADARFT (denoted as "w/o ADARFT") as a baseline.

As shown in Figure 4, the model trained with $\beta = 0.5$ achieves the highest accuracy throughout training. This supports our theoretical motivation in Section 3.4: maximizing reward variance, which occurs when success rate $\approx 0.5$, provides the strongest learning signal. Models with $\beta = 0.2$ and $\beta = 0.8$ underperform likely due to curriculum misalignment: $\beta = 0.8$ overly focuses on easy problems, while $\beta = 0.2$ overemphasizes difficult ones, both of which limit the model's capacity to generalize. The reward and difficulty curves align with the accuracy outcomes discussed above. The $\beta = 0.5$ configuration maintains a stable reward near 0.5, reflecting balanced difficulty exposure. In contrast, $\beta = 0.8$ results in overly high reward (i.e., easy samples), while $\beta = 0.2$ maintains a reward around 0.2 for most of training, indicating the model is repeatedly presented with overly difficult problems. As expected, response length is the shortest for $\beta = 0.8$ and longest for $\beta = 0.2$, consistent with the idea that longer responses correlate with problem complexity.

### 5.4 Data Difficulty on Model Performance

To better understand the effect of data difficulty on model performance, we introduce two additional data distributions: easy-extreme and hard-extreme. Unlike the skew-difficult and skew-easy distributions, which still include a mix of difficulty levels, the easy-extreme and hard-extreme sets consist exclusively of the most polarized examples. Specifically, easy-extreme contains only the easiest samples with difficulty levels no greater than 15, while hard-extreme includes only the hardest samples with difficulty levels of at least 97. Each of these extreme distributions consists of approximately 8,000 samples, providing a focused and controlled evaluation of model behavior under minimal or maximal difficulty conditions. We trained a Qwen 2.5 7B model on each of the two extreme distributions using PPO, and compared their performance to models trained on the uniform distribution with PPO (Uniform) and with ADARFT instantiated with PPO (Uniform + ADARFT), as described in Section 5. The results are presented in Figure 5. The key takeaway is that training on only overly easy or hard problems fails to provide useful learning signals, reinforcing the need for ADARFT to adaptively steer models toward challenges matched to their current ability.

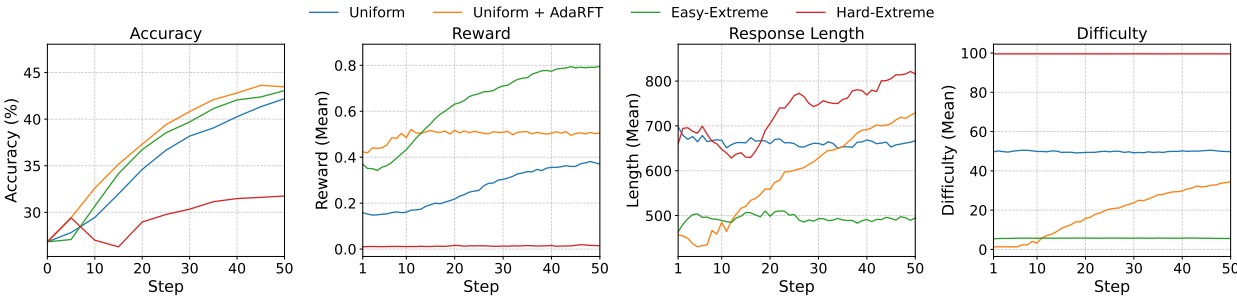

Figure 5: Performance comparison of Qwen 2.5 7B trained on different data distributions using PPO (Uniform, Easy-Extreme, Hard-Extreme) and ADARFT instantiated with PPO (Uniform + ADARFT). For clarity, curves are exponentially smoothed ($\alpha = 0.3$) to reduce noise.

**Accuracy.** The leftmost panel of Figure 5 shows that uniform + ADARFT achieves the highest overall accuracy throughout training, outperforming both uniform and the two extreme settings. This highlights the effectiveness of ADARFT in guiding the model through an optimal difficulty progression. In contrast, hard-extreme struggles, with a flat and lower trajectory, indicating that exposing the model only to very

difficult problems limits learning progress. This suggests that without a gradual exposure strategy, models trained on only the hardest problems are unable to bootstrap their capabilities effectively.

**Reward.** The reward trends provide important clues about learning dynamics. The easy-extreme setup achieves the fastest reward improvement during early training, surpassing both uniform and hard-extreme. In particular, easy-extreme consistently operates in a reward range between 0.4 and 0.6 during early training, which corresponds to a success rate that is both challenging and attainable. In contrast, the reward of the uniform and hard-extreme setup lingers below 0.2 in early training, leading to slower learning. This suggests that training on problems with intermediate difficulty—those that are neither trivially easy nor prohibitively hard—provides the most effective learning signal. Notably, ADARFT is explicitly designed to exploit this insight: by setting the target reward $\beta = 0.5$, we encourage the model to train on problems that match this "productive struggle" zone. As shown by the uniform + ADARFT curve, the algorithm successfully maintains an average reward near 0.5 throughout training, allowing the model to learn at an optimal pace. Notably, while the uniform setup eventually reaches a reward of nearly 0.5 by step 50, it does not result in faster learning. This is likely because the model is already fairly well trained by that stage, so the additional reward signal contributes less to further improvement. In contrast, the hard-extreme model receives almost no reward signal for most of the training, while the uniform setup shows slower and more gradual reward accumulation.

**Response Length.** The response length panel reveals how the complexity of generated solutions evolves during training. The hard-extreme model consistently produces the longest responses, with length increasing steadily, reflecting the higher complexity and reasoning depth required by the hardest problems. In contrast, the easy-extreme setup maintains short and stable responses, consistent with its simpler problem set. The uniform and uniform + ADARFT setups fall between these two extremes. Notably, uniform + ADARFT shows a gradual increase in response length over time. This trend aligns with the behavior of the curriculum learning algorithm: as the model improves, it is exposed to increasingly difficult problems, which naturally demand more elaborate reasoning and longer solutions. This dynamic suggests that response length can serve as a useful proxy for problem difficulty and reasoning complexity during training.

**Difficulty.** Finally, the difficulty panel illustrates how problem difficulty evolves under each setup. The easy-extreme and hard-extreme curves remain flat, confirming that these datasets contain only problems from the tail ends of the difficulty spectrum (i.e., $\leq 15$ and $\geq 97$, respectively). The uniform curve is centered around 50, as expected, while uniform + ADARFT shows a steady increase in difficulty over time. This adaptive progression confirms that curriculum learning effectively steers the model from easier to harder problems, aligning difficulty with the model's evolving capabilities.

### 5.5 AdaRFT with Diverse RL Algorithms

To evaluate the generality of ADARFT beyond PPO, we trained the Qwen 2.5 Math 1.5B model on a skew-difficult data distribution using two alternative reinforcement learning algorithms: REINFORCE++ and GRPO (implementation details in Appendix B). As shown in Figure 6, ADARFT improves convergence speed and final accuracy across these variants in our single-seed runs. Across both cases, the adaptive curriculum acts orthogonally to the underlying optimization method. These results reinforce the plug-and-play nature of ADARFT: it consistently enhances sample efficiency across algorithmic choices, making it broadly applicable in diverse reinforcement finetuning pipelines. Notably, this generalization holds without any additional tuning or algorithm-specific modifications, underscoring the practical utility of curriculum-aware training in both lightweight and computation-heavy RFT settings.

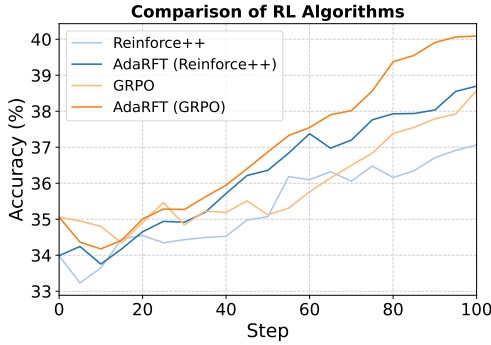

Figure 6: Comparison between models trained with and without AdaRFT using REINFORCE++ and GRPO.

### 5.6 Training on LLM-Estimated Difficulty

In addition to rollout-based difficulty estimation, we explore an alternative strategy that uses LLM-judged difficulty levels to guide curriculum construction. As described in Section 4.1, we prompt GPT-4o (`gpt-4o-0806`) to assign difficulty levels to math problems in the DeepScaleR dataset according to the AoPS rubric. This approach offers a lightweight and scalable alternative to computing pass@$k$ success rates from model rollouts, making it especially attractive in low-resource scenarios.

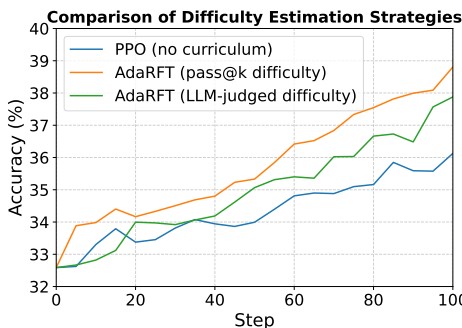

Figure 7: Comparison of different difficulty estimation strategies.

To assess the effectiveness of this strategy, we train a Qwen 2.5 Math 1.5B model on the skew-difficult distribution using ADARFT (PPO) with two curriculum schedules: one based on rollout-derived pass@$k$ difficulty, and the other guided by GPT-4o's difficulty ratings. Since the LLM-judged difficulty is on a scale of 1 to 5 (rather than 0 to 100), we set the step size hyperparameter $\eta = 2.5$ to align the difficulty adjustment magnitude with the reward signal. All other hyperparameters are kept unchanged. As shown in Figure 7, both curriculum strategies outperform standard PPO without curriculum learning. While rollout-based difficulty estimation yields the strongest gains, the LLM-judged curriculum still provides a noticeable improvement over the baseline.

These results demonstrate that ADARFT remains effective even when the difficulty signal is derived from heuristic or approximate sources like LLM judgments. Although less precise than empirical pass@$k$ metrics, the LLM-based difficulty still provides enough structure to enable meaningful curriculum adaptation. This makes it a practical fallback when rollout computation is too costly, and suggests that future work could explore hybrid approaches that combine lightweight heuristics with periodic empirical calibration.

## 6 Discussion

**When Does Curriculum Learning Help Most?** Our findings show that curriculum learning provides the greatest benefits under two key conditions: (1) imbalanced training distributions, and (2) limited model capacity. In skewed settings, standard PPO often struggles early due to sparse reward signals. ADARFT mitigates this issue by prioritizing easier problems, allowing the model to bootstrap learning before progressing to harder ones. The gains are smaller when the model is strong or the data is well-balanced, since the training distribution is already sufficiently informative. Manual data curation and fixed curriculum schedules can partially address these issues (Yu et al., 2025; Shen et al., 2025; Chen et al., 2025; Zeng et al., 2025), but they require substantial human effort and domain expertise and often need to be re-tuned for each model or task. Fixed curricula also cannot adapt to differences in learning speed. We provide a detailed analysis of fixed curricula in Appendix A.1. In contrast, ADARFT dynamically adjusts difficulty based on the model's reward signal without manual intervention. This makes it broadly applicable across model scales and data distributions, improving scalability and robustness, especially in fixed data settings where it can adapt to the model without modifying the dataset.

**Use of Offline Difficulty Estimation.** A core design choice of ADARFT is to rely on offline difficulty estimates that remain fixed throughout training, rather than recomputing difficulty online as the model improves. ADARFT adopts a solver-independent notion of difficulty, in which problems such as IMO-level questions are considered inherently more difficult than grade-school math problems and are assigned a single difficulty score based on intrinsic complexity, in contrast to online curriculum methods that use solver-dependent definitions that evolve with the model's success rate. This design aligns naturally with many mathematics competitions (Hendrycks et al., 2021) and puzzle datasets (Lin et al., 2025), which provide fixed difficulty annotations, and with modern RL fine-tuning pipelines where pass@$n$ rollouts are commonly computed during data preparation regardless of curriculum learning. Recent systems such as Open Reasoner Zero (Hu et al., 2025), Kimi k1.5 (Kimi et al., 2025), and ZR1 1.5B (Zyphra, 2025) evaluate pass@$n$ on the full dataset prior to training to filter overly easy or difficult examples, making difficulty information readily available in practice. Rather than updating difficulty estimates themselves, ADARFT adapts the curriculum by dynamically updating the target difficulty $T$ based on the model's success rate on the currently

sampled data, achieving the same functional outcome as online difficulty estimation while remaining stable and scalable for large-scale RL fine-tuning.

We report curriculum construction and policy optimization as separate sources of compute. The efficiency results in Table 1 measure the RFT training phase after the curriculum scores have been assigned; they do not include the upfront difficulty-estimation pass. In our main rollout-based setup, assigning difficulty to a 10,000-example training pool with pass@128 requires 1.28M scorer rollouts. This cost should be included explicitly for one-off training runs, and our $2\times$ speedup claim should therefore be interpreted as a reduction in the subsequent RFT training phase. The cost can be reduced by using fewer rollouts, since Figure 1a shows that pass@40 and pass@64 preserve much of the pass@128 ordering, or by using LLM-judged difficulty as in Section 5.6. It can also be amortized when the same scored dataset is reused across model sizes, algorithms, hyperparameter sweeps, or future training runs.

## 7    Conclusion

We propose ADARFT, an adaptive curriculum learning strategy for reinforcement finetuning (RFT) that dynamically matches problem difficulty to a model's evolving skill level. By adjusting a target difficulty based on reward feedback, ADARFT improves sample and compute efficiency during RFT training with problem-level difficulty annotations, without modifying the reward function or underlying RL algorithm. Experiments across multiple data regimes and model sizes show consistent gains in convergence speed, especially in imbalanced training distributions. This lightweight, scalable approach highlights the value of curriculum-aware training for efficient and robust alignment in structured reasoning tasks.

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

# A   Appendix

## A.1   Training Dynamics: AdaRFT vs. Fixed Curriculum vs. PPO

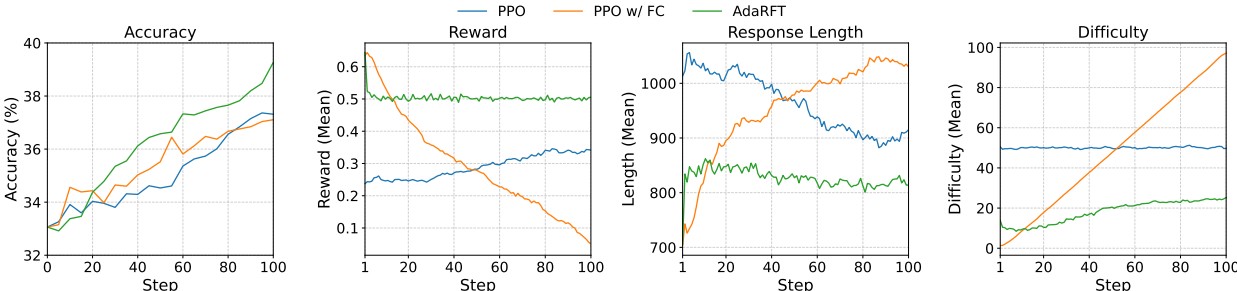

Figure 8: Training dynamics of standard PPO, PPO with a fixed curriculum (PPO w/ FC), and ADARFT on the Qwen 2.5 Math 1.5B model under the uniform data distribution. We plot accuracy, reward, response length, and average difficulty of sampled training problems over training steps. Curves are exponentially smoothed ($\alpha = 0.3$) for clarity.

To further contextualize the role of adaptive curricula, we analyze the training dynamics of fixed curriculum (PPO w/ FC), standard PPO, and ADARFT when training the Qwen 2.5 Math 1.5B model on the uniform distribution. Figure 8 illustrates the evolution of accuracy, reward, response length, and sampled difficulty across the first 100 training steps.

The accuracy curves highlight the central tradeoff of fixed curricula. PPO w/ FC initially converges slightly faster than standard PPO, benefiting from early exposure to easier problems. However, it ultimately underperforms compared to ADARFT because its difficulty schedule increases independently of the model's actual learning progress. As a result, the model is pushed into harder problem regimes too quickly. This misalignment is clearly visible in the reward curve. Early in training, PPO w/ FC achieves rewards well above 0.5, indicating that the model is initially exposed to problems that are easier than its current capability. However, as training progresses, the fixed curriculum increases difficulty at a rate that outpaces the model's learning speed. As a result, the reward drops below 0.5 and continues declining, showing that the model is increasingly confronted with problems it cannot yet solve. This mismatch limits the effectiveness of the updates and ultimately leads to slower convergence compared to an adaptive curriculum.

The response length patterns reinforce this interpretation. Because longer responses typically correspond to more difficult reasoning tasks, the rapid increase in response length under PPO w/ FC shows that the curriculum escalates difficulty faster than the model can adapt. In contrast, standard PPO maintains more stable lengths but lacks the structured progression necessary for efficient learning. ADARFT, by comparison, keeps response lengths moderate and gradually increasing, consistent with its difficulty traces: the curriculum raises problem difficulty only when the model's reward stays near the target value, ensuring that the model always receives examples that are challenging but solvable.

This particular dynamic suggest a general underlying issue: since the model's perceived difficulty changes over the course of training, a fixed curriculum cannot remain aligned, and different model-dataset combinations may experience extended periods of being over-challenged or under-challenged. This mismatch limits the effectiveness of updates and leads to slower convergence compared to an adaptive curriculum. In contrast, ADARFT is designed to handle precisely this challenge: by adjusting the difficulty schedule based on the model's reward signal, it continually matches training difficulty to the model's evolving capability, ensuring sustained learning progress throughout training. Without the ability to adjust to real-time model performance, fixed schedules risk either overwhelming the model or wasting compute on overly easy tasks. ADARFT avoids both extremes by maintaining the average reward near 0.5, automatically pacing the introduction of harder problems as the model improves. This adaptive alignment between task difficulty and model capability leads to smoother reward trajectories, controlled response lengths, and ultimately more stable and efficient learning.

| Model | Setup | Method | GSM8K | MATH 500 | Olympiad Bench | Minerva Math | AMC 23 (Avg@8) | AIME 24 (Avg@8) | Average |
|---|---|---|---|---|---|---|---|---|---|
| Qwen 2.5 Math 1.5B | skew-difficult | ADARFT | **74.00** | **66.40** | **20.36** | **15.07** | **55.00** | **12.08** | **40.48** |
| | | PPO | 69.67 | 64.60 | 20.65 | 12.87 | 47.50 | 9.17 | 37.41 |
| | | PPO (w/ Filter) | 71.65 | 62.40 | 20.06 | **15.07** | 45.00 | 9.17 | 37.22 |
| | | PPO (w/ FC) | 72.55 | 66.40 | 20.95 | 14.34 | 45.00 | 4.06 | 37.22 |
| | uniform | ADARFT | **74.53** | **66.20** | **21.99** | 14.34 | **57.50** | **12.08** | **41.11** |
| | | PPO | 71.95 | 65.20 | 21.10 | **15.81** | 42.50 | 6.67 | 37.20 |
| | | PPO (w/ Filter) | 72.63 | 65.80 | 20.21 | 13.60 | 45.00 | 10.00 | 37.87 |
| | | PPO (w/ FC) | 71.95 | 65.60 | 19.61 | 14.71 | 42.50 | 9.27 | 37.27 |
| | skew-easy | ADARFT | 73.62 | 66.20 | 19.91 | **13.97** | **55.00** | 9.17 | **39.18** |
| | | PPO | 72.71 | **67.40** | 19.17 | **13.97** | 45.00 | **12.50** | 38.46 |
| | | PPO (w/ Filter) | **74.75** | 65.20 | **20.36** | 13.60 | 45.00 | 10.00 | 38.15 |
| | | PPO (w/ FC) | 72.40 | 66.20 | 20.06 | 13.24 | 50.00 | 6.67 | 38.09 |
| Qwen 2.5 7B | skew-difficult | ADARFT | **90.98** | 71.40 | **25.85** | 22.43 | **52.50** | **15.83** | **46.83** |
| | | PPO | 89.69 | 71.20 | 23.33 | 23.53 | 50.00 | 11.25 | 44.17 |
| | | PPO (w/ Filter) | 88.48 | **72.20** | 24.37 | 25.00 | 50.00 | 12.08 | 45.35 |
| | | PPO (w/ FC) | 89.92 | 72.00 | 24.52 | 25.37 | 47.50 | 13.96 | 45.54 |
| | uniform | ADARFT | **90.14** | 72.60 | **24.96** | 24.26 | **55.00** | 14.58 | **46.92** |
| | | PPO | 89.31 | 72.40 | 23.63 | **25.37** | 42.50 | **15.00** | 44.70 |
| | | PPO (w/ Filter) | 89.08 | **74.40** | 23.18 | 22.43 | 45.00 | 13.33 | 44.57 |
| | | PPO (w/ FC) | 89.84 | 72.40 | 23.92 | 23.90 | 42.50 | 13.33 | 44.32 |
| | skew-easy | ADARFT | **90.14** | 72.60 | **25.56** | 23.16 | **50.00** | **14.17** | **45.94** |
| | | PPO | 89.39 | **73.60** | 23.33 | **24.26** | 47.50 | 13.33 | 45.07 |
| | | PPO (w/ Filter) | 89.31 | 71.60 | 24.22 | 23.90 | 47.50 | 13.33 | 44.98 |
| | | PPO (w/ FC) | 89.46 | 72.20 | 24.37 | 23.90 | 37.50 | 13.33 | 43.46 |

Table 2: Accuracy (%) at step 100 for every model, setup, and benchmark. ADARFT in this table refers to ADARFT instantiated with PPO, i.e., ADARFT (PPO).

## A.2 Detailed Breakdown of Model Performance

Table 2 provides a per-benchmark breakdown of final accuracy at step 100 across all models, data setups, and training methods. Results are reported on six diverse mathematical reasoning benchmarks spanning grade-school arithmetic (GSM8K), competition-style problems (MATH 500, AMC 23, AIME 24), and advanced proof-oriented datasets (OlympiadBench, Minerva Math). Across both model scales and all data distributions, ADARFT either matches or outperforms PPO-based baselines on the majority of individual benchmarks, resulting in consistent gains in the overall average accuracy.

For Qwen 2.5 Math 1.5B, improvements from ADARFT are particularly pronounced on GSM8K, MATH 500, AMC 23, and AIME 24 under skew-difficult and uniform settings, indicating stronger generalization across both structured and competition-style problems. Even in the skew-easy setting, where baseline PPO already performs well, ADARFT maintains or slightly improves performance without introducing regressions on harder benchmarks such as OlympiadBench and Minerva Math. For the larger Qwen 2.5 7B model, gains are more modest but remain consistent, with ADARFT achieving higher average accuracy across all three data distributions and stable performance on the most challenging benchmarks. Overall, these results confirm that the efficiency gains of ADARFT do not come at the cost of final model quality, and in practice often translate into improved performance across a broad range of mathematical reasoning tasks.

# B Implementation Details

## B.1 Training Configuration

We trained both the actor and critic models using the PPO algorithm on a single node with 8 A100 GPUs. Each model was trained for approximately 100 optimization steps using the `veRL` library (Sheng et al., 2024). We used two model variants: `Qwen2.5-7B` and `Qwen2.5-MATH-1.5B`. The latter has a shorted context window, so we adjusted the max response length and the sequence parallel size accordingly.

Table 3 summarizes the core hyperparameter settings used across all three algorithms: PPO, GRPO, and REINFORCE++. We highlight both shared defaults and algorithm-specific overrides, including KL treatment modes, rollout settings, and critic configurations.

## B.2 Deriving the Target-Difficulty Update Rule from a Linear Mapping

A central component of our curriculum mechanism is the update of the target difficulty $T$ based on the model's observed reward performance. While the final update rule (Eq. 13) involves a hyperbolic tangent, it is in fact a smooth and stabilized variant of a standard linear mapping between reward space and difficulty space. We derive it here for clarity.

**Step 1: Linear mapping between two intervals.** The classical linear interpolation formula for mapping a value $v \in [x, y]$ to a target interval $[a, b]$ is

$$v' = a + \frac{(v - x)(b - a)}{y - x}. \tag{5}$$

If we directly map the average reward $R_{\text{avg}} \in [r_{\min}, r_{\max}]$ to the difficulty range $[d_{\min}, d_{\max}]$, we obtain

$$T_{\text{naive}}(R_{\text{avg}}) = d_{\min} + \frac{(R_{\text{avg}} - r_{\min})(d_{\max} - d_{\min})}{r_{\max} - r_{\min}}. \tag{6}$$

In our main setting, $r_{\min} = 0$, $r_{\max} = 1$, and $[d_{\min}, d_{\max}] = [0, 100]$, so the naive mapping simplifies to

$$T_{\text{naive}}(R_{\text{avg}}) = 100 \, R_{\text{avg}}. \tag{7}$$

**Step 2: Mapping reward deviation instead of absolute reward.** For curriculum learning, we do not wish to reassign a new difficulty level at every step. Instead, we aim to *adjust* the current target difficulty depending on whether the model is performing above or below a desired target success rate $\beta$. We therefore consider the deviation

$$\delta = R_{\text{avg}} - \beta. \tag{8}$$

Given $R_{\text{avg}} \in [0, 1]$, the deviation satisfies $\delta \in [r_{\min} - \beta, \, r_{\max} - \beta] = [-\beta, \, 1 - \beta]$. With the common choice $\beta = 0.5$, this becomes $\delta \in [-0.5, 0.5]$.

Applying the linear mapping rule equation 5 from the deviation range $[-0.5, 0.5]$ to a symmetric difficulty-change interval $[-\Delta, \Delta]$ yields

$$\begin{aligned} \Delta T_{\text{lin}}(\delta) &= -\Delta + \frac{(\delta - (-0.5))(\Delta - (-\Delta))}{0.5 - (-0.5)} \\ &= 2\Delta \, \delta. \end{aligned} \tag{9}$$

Thus the naive linear controller becomes

$$T'_{\text{lin}} = T + 2\Delta \, (R_{\text{avg}} - \beta). \tag{10}$$

This already captures the desired behavior: difficulty increases when performance exceeds the target, decreases when performance falls short, and remains stable when $R_{\text{avg}} = \beta$.

**Step 3: Stabilizing the update via a smooth saturating nonlinearity.** A purely linear controller may cause excessively large changes when the reward deviation is large or noisy. To obtain a stable update rule, we replace the linear term with a smooth, odd, saturating nonlinearity. The hyperbolic tangent is a natural choice: it behaves linearly near zero (which recovers the linear mapping) and saturates as its argument grows.

We therefore define a smoothed difficulty adjustment

$$\Delta T(\delta) = \eta \cdot \tanh\big(\alpha(R_{\mathrm{avg}} - \beta)\big). \tag{11}$$

Here, $\eta$ sets the maximum update magnitude and $\alpha$ controls the sensitivity around the target reward. For small deviations, $\tanh(z) \approx z$, so locally

$$\Delta T(\delta) \approx \eta\alpha(R_{\mathrm{avg}} - \beta), \tag{12}$$

recovering a linear controller with effective slope $\eta\alpha$ while ensuring global boundedness.

**Step 4: Clipping to the valid difficulty range.** To ensure the target difficulty remains within the observed range of the data, we apply a final clipping:

$$T' = \mathrm{clip}\big(T + \eta \cdot \tanh\big(\alpha(R_{\mathrm{avg}} - \beta)\big),\ d_{\min},\ d_{\max}\big). \tag{13}$$

The full update rule equation 13 is therefore a direct, smoothed generalization of a naive linear mapping between reward deviations and difficulty adjustments. It preserves the intuitive behavior of the linear controller near the target reward, while the saturating nonlinearity and clipping ensure stable, bounded, and data-consistent curriculum updates.

Because the reward is bounded in $[0, 1]$ and the difficulty metric spans $[0, 100]$, we set the step size $\eta = 50$ to align their scales. The modulation parameter $\alpha = 2$ ensures smooth and controlled progression throughout training.

### B.3 Prompt for Difficulty Estimation Using LLM as a Judge

The prompt used for difficulty estimation (as described in Section 4.1) is shown in Table 4, Table 5, and Table 6. The descriptions of the difficulty scales and examples are sourced from the AoPS Wiki.[1] Although GPT-4o is prompted to rate problem difficulty on a scale from 1 to 10, we found that over 95% of the problems fall within the range of 1 to 5. Therefore, we clip the scores and use a revised scale from 1 to 5. In addition to integer scores, we also allow half-point increments such as 1.5, 2.0, and 2.5 for finer-grained difficulty estimation.

---

[1]https://artofproblemsolving.com/wiki/index.php/AoPS_Wiki:Competition_ratings

| Category | Parameter | PPO | GRPO | REINFORCE++ |
|---|---|---|---|---|
| *Algorithm-Specific Settings* | | | | |
| General | Advantage estimator | GAE | GRPO | REINFORCE++ |
| | Gamma ($\gamma$) | 1.0 | — | — |
| | Lambda ($\lambda$) | 1.0 | — | — |
| | Batch size | 1024 | 1024 | 1024 |
| | Max prompt length | 1024 | 1024 | 1024 |
| | Gradient checkpointing | Enabled | Enabled | Enabled |
| Actor | Learning rate | $1 \times 10^{-6}$ | $1 \times 10^{-6}$ | $1 \times 10^{-6}$ |
| | Mini-batch size | 1024 | 1024 | 1024 |
| | Dynamic batch size | Enabled | Enabled | Enabled |
| | KL penalty role | Reward | Loss | Loss |
| | KL loss type | Fixed | Low-variance KL | MSE |
| | KL loss coefficient ($\beta$) | 0.001 | 0.001 | 0.001 |
| | Entropy coefficient | 0.001 | 0.001 | 0 |
| | Clip ratio | 0.2 | 0.2 | 0.2 |
| | Gradient clipping | 1.0 | 1.0 | 1.0 |
| | Sequence parallel size | Model-specific | Model-specific | Model-specific |
| Rollout | Backend | vLLM | vLLM | vLLM |
| | Tensor model parallel size | 2 | 2 | 2 |
| | Rollouts per sample | 1 | 8 | 1 |
| | Nucleus sampling $p$ | 1.0 | 1.0 | 1.0 |
| | GPU memory utilization | 0.5 | 0.5 | 0.5 |
| | Sampling temperature | 1.0 | 1.0 | 1.0 |
| Critic | Warmup steps | 0 | — | — |
| | Learning rate | $1 \times 10^{-5}$ | — | — |
| | Sequence parallel size | Model-specific | — | — |
| *Model-Specific Overrides (shared across all algorithms)* | | | | |
| Qwen2.5-7B | Max response length | 8000 | 8000 | 8000 |
| | Sequence parallel size | 2 | 2 | 2 |
| | Max token length / GPU | 8000 | 8000 | 8000 |
| Qwen2.5-MATH-1.5B | Max response length | 3000 | 3000 | 3000 |
| | Sequence parallel size | 1 | 1 | 1 |
| | Max token length / GPU | 16000 | 16000 | 16000 |
| *ADARFT Parameters* | | | | |
| Curriculum Learning | Target reward ($\beta$) | 0.5 | 0.5 | 0.5 |
| | Sensitivity ($\alpha$) | 2 | 2 | 2 |
| | Step size ($\eta$) | 50 | 50 | 50 |
| | Initial difficulty ($T$) | 0 | 0 | 0 |

Table 3: Comparison of training hyperparameters for PPO, GRPO, and REINFORCE++ using the veRL library. Shared defaults are used unless overridden.

---

**Prompt for Difficulty Estimation (Part 1)**

- - - - - - - - - - - - - - - - - - - - - - - - - - - - - - - - - - - - - - - - - - - - - - - - - - -

# Math Problem
{problem}

# Your Task
You are a subject matter expert in mathematics tasked with evaluating the difficulty level of individual math problems. Your assessment should be objective and based on a detailed difficulty scale provided below. Your judgment will help calibrate and categorize problems for use in educational settings or assessments. Be thorough, fair, and consistent in your evaluation.

# Difficulty Scale
1: Problems strictly for beginner, on the easiest elementary school or middle school levels (MOEMS, MATHCOUNTS School, AMC 8 1-10, AMC 10 1-10, easier AMC 12 1-5, and others that involve standard techniques introduced up to the middle school level), most traditional middle/high school word problems.
1.5: Problems for stronger beginner students, on the level of the middling problems in most middle school contests (AMC 8 11-20, harder AMC 10 1-10, AMC 12 1-5, and those others that force students to apply their school-level knowledge to slightly more challenging problems), traditional middle/high school word problems with more complex problem solving.
2: For motivated beginners, harder questions from the previous categories (AMC 8 21-25, MATH-COUNTS Chapter (Sprint 21-30, Target 6-8), MATHCOUNTS States/Nationals, AMC 10 11-15, AMC 12 5-10, easiest AIME 1-3)
2.5: More advanced beginner problems, hardest questions from previous categories (Harder AMC 8 21-25, harder MATHCOUNTS States questions, AMC 10 16-20, AMC 12 11-15, usual AIME 1-3)
3: Early intermediate problems that require more creative thinking (harder MATHCOUNTS National questions, AMC 10 21-25, AMC 12 15-20, hardest AIME 1-3, usual AIME 4-6).
4: Intermediate-level problems (AMC 12 21-25, hardest AIME 4-6, usual AIME 7-10).
5: More difficult AIME problems (11-13), simple proof-based Olympiad-style problems (early JBMO questions, easiest USAJMO 1/4).
6: High-leveled AIME-styled questions (14/15). Introductory-leveled Olympiad-level questions (harder USAJMO 1/4 and easier USAJMO 2/5, easier USAMO and IMO 1/4).
7: Tougher Olympiad-level questions, may require more technical knowledge (harder USAJMO 2/5 and most USAJMO 3/6, extremely hard USAMO and IMO 1/4, easy-medium USAMO and IMO 2/5).
8: High-level Olympiad-level questions (medium-hard USAMO and IMO 2/5, easiest USAMO and IMO 3/6).
9: Expert Olympiad-level questions (average USAMO and IMO 3/6).
9.5: The hardest problems appearing on Olympiads which the strongest students could reasonably solve (hard USAMO and IMO 3/6).
10: Historically hard problems, generally unsuitable for very hard competitions (such as the IMO) due to being exceedingly tedious, long, and difficult (e.g. very few students are capable of solving on a worldwide basis).

Table 4: Prompt for difficulty estimation using LLM as a judge.

**Prompt for Difficulty Estimation (Part 2)**

- - - - - - - - - - - - - - - - - - - - - - - - - - - - - - - - - - - - - - - - - - - - - - -

# Examples
For reference, here are some sample problems from each of the difficulty levels 1-10:

<1: Jamie counted the number of edges of a cube, Jimmy counted the numbers of corners, and Judy counted the number of faces. They then added the three numbers. What was the resulting sum? (2003 AMC 8, Problem 1)

1: How many integer values of $x$ satisfy $|x| < 3\pi$? (2021 Spring AMC 10B, Problem 1)

1.5: A number is called flippy if its digits alternate between two distinct digits. For example, 2020 and 37373 are flippy, but 3883 and 123123 are not. How many five-digit flippy numbers are divisible by 15? (2020 AMC 8, Problem 19)

2: A fair 6-sided die is repeatedly rolled until an odd number appears. What is the probability that every even number appears at least once before the first occurrence of an odd number? (2021 Spring AMC 10B, Problem 18)

2.5: $A$, $B$, $C$ are three piles of rocks. The mean weight of the rocks in $A$ is 40 pounds, the mean weight of the rocks in $B$ is 50 pounds, the mean weight of the rocks in the combined piles $A$ and $B$ is 43 pounds, and the mean weight of the rocks in the combined piles $A$ and $C$ is 44 pounds. What is the greatest possible integer value for the mean in pounds of the rocks in the combined piles $B$ and $C$? (2013 AMC 12A, Problem 16)

3: Triangle $ABC$ with $AB = 50$ and $AC = 10$ has area 120. Let $D$ be the midpoint of $\overline{AB}$, and let $E$ be the midpoint of $\overline{AC}$. The angle bisector of $\angle BAC$ intersects $\overline{DE}$ and $\overline{BC}$ at $F$ and $G$, respectively. What is the area of quadrilateral $FDBG$? (2018 AMC 10A, Problem 24)

3.5: Find the number of integer values of $k$ in the closed interval $[-500, 500]$ for which the equation $\log(kx) = 2\log(x+2)$ has exactly one real solution. (2017 AIME II, Problem 7)

4: Define a sequence recursively by $x_0 = 5$ and

$$x_{n+1} = \frac{x_n^2 + 5x_n + 4}{x_n + 6}$$

for all nonnegative integers $n$. Let $m$ be the least positive integer such that

$$x_m \leq 4 + \frac{1}{2^{20}}.$$

In which of the following intervals does $m$ lie?
**(A)** $[9, 26]$    **(B)** $[27, 80]$    **(C)** $[81, 242]$    **(D)** $[243, 728]$    **(E)** $[729, \infty)$ (2019 AMC 10B, Problem 24 and 2019 AMC 12B, Problem 22)

Table 5: Prompt for difficulty estimation using LLM as a judge.

**Prompt for Difficulty Estimation (Part 3)**

- - - - - - - - - - - - - - - - - - - - - - - - - - - - - - - - - - - - - - - - - - - - - - - - - - - - - - - - - - - - - - -

4.5: Find, with proof, all positive integers $n$ for which $2^n + 12^n + 2011^n$ is a perfect square. (USAJMO 2011/1)

5: Find all triples $(a, b, c)$ of real numbers such that the following system holds:

$$a + b + c = \frac{1}{a} + \frac{1}{b} + \frac{1}{c},$$

$$a^2 + b^2 + c^2 = \frac{1}{a^2} + \frac{1}{b^2} + \frac{1}{c^2}.$$

(JBMO 2020/1)

5.5: Triangle $ABC$ has $\angle BAC = 60°$, $\angle CBA \leq 90°$, $BC = 1$, and $AC \geq AB$. Let $H$, $I$, and $O$ be the orthocenter, incenter, and circumcenter of $\triangle ABC$, respectively. Assume that the area of pentagon $BCOIH$ is the maximum possible. What is $\angle CBA$? (2011 AMC 12A, Problem 25)

6: Let $\triangle ABC$ be an acute triangle with circumcircle $\omega$, and let $H$ be the intersection of the altitudes of $\triangle ABC$. Suppose the tangent to the circumcircle of $\triangle HBC$ at $H$ intersects $\omega$ at points $X$ and $Y$ with $HA = 3, HX = 2$, and $HY = 6$. The area of $\triangle ABC$ can be written in the form $m\sqrt{n}$, where $m$ and $n$ are positive integers, and $n$ is not divisible by the square of any prime. Find $m + n$. (2020 AIME I, Problem 15)

6.5: Rectangles $BCC_1B_2$, $CAA_1C_2$, and $ABB_1A_2$ are erected outside an acute triangle $ABC$. Suppose that

$$\angle BC_1C + \angle CA_1A + \angle AB_1B = 180°.$$

Prove that lines $B_1C_2$, $C_1A_2$, and $A_1B_2$ are concurrent. (USAMO 2021/1, USAJMO 2021/2)

7: We say that a finite set $\mathcal{S}$ in the plane is balanced if, for any two different points $A$, $B$ in $\mathcal{S}$, there is a point $C$ in $\mathcal{S}$ such that $AC = BC$. We say that $\mathcal{S}$ is centre-free if for any three points $A$, $B$, $C$ in $\mathcal{S}$, there is no point $P$ in $\mathcal{S}$ such that $PA = PB = PC$.

Show that for all integers $n \geq 3$, there exists a balanced set consisting of $n$ points. Determine all integers $n \geq 3$ for which there exists a balanced centre-free set consisting of $n$ points. (IMO 2015/1)

7.5: Let $\mathbb{Z}$ be the set of integers. Find all functions $f : \mathbb{Z} \to \mathbb{Z}$ such that

$$xf(2f(y) - x) + y^2f(2x - f(y)) = \frac{f(x)^2}{x} + f(yf(y))$$

for all $x, y \in \mathbb{Z}$ with $x \neq 0$. (USAMO 2014/2)

Table 6: Prompt for difficulty estimation using LLM as a judge.

---

**Prompt for Difficulty Estimation (Part 4)**

- - - - - - - - - - - - - - - - - - - - - - - - - - - - - - - - - - - - - - - - - - - - - - - - - -

8: For each positive integer $n$, the Bank of Cape Town issues coins of denomination $\frac{1}{n}$. Given a finite collection of such coins (of not necessarily different denominations) with total value at most most $99 + \frac{1}{2}$, prove that it is possible to split this collection into 100 or fewer groups, such that each group has total value at most 1. (IMO 2014/5)

8.5: Let $I$ be the incentre of acute triangle $ABC$ with $AB \neq AC$. The incircle $\omega$ of $ABC$ is tangent to sides $BC, CA$, and $AB$ at $D, E$, and $F$, respectively. The line through $D$ perpendicular to $EF$ meets $\omega$ at $R$. Line $AR$ meets $\omega$ again at $P$. The circumcircles of triangle $PCE$ and $PBF$ meet again at $Q$. Prove that lines $DI$ and $PQ$ meet on the line through $A$ perpendicular to $AI$. (IMO 2019/6)

9: Let $k$ be a positive integer and let $S$ be a finite set of odd prime numbers. Prove that there is at most one way (up to rotation and reflection) to place the elements of $S$ around the circle such that the product of any two neighbors is of the form $x^2 + x + k$ for some positive integer $x$. (IMO 2022/3)

9.5: An anti-Pascal triangle is an equilateral triangular array of numbers such that, except for the numbers in the bottom row, each number is the absolute value of the difference of the two numbers immediately below it. For example, the following is an anti-Pascal triangle with four rows which contains every integer from 1 to 10.

$$4$$
$$2 \quad 6$$
$$5 \quad 7 \quad 1$$
$$8 \quad 3 \quad 10 \quad 9$$

Does there exist an anti-Pascal triangle with 2018 rows which contains every integer from 1 to $1 + 2 + 3 + \cdots + 2018$? (IMO 2018/3)

10: Prove that there exists a positive constant $c$ such that the following statement is true: Consider an integer $n > 1$, and a set $\mathcal{S}$ of $n$ points in the plane such that the distance between any two different points in $\mathcal{S}$ is at least 1. It follows that there is a line $\ell$ separating $\mathcal{S}$ such that the distance from any point of $\mathcal{S}$ to $\ell$ is at least $cn^{-1/3}$.

(A line $\ell$ separates a set of points S if some segment joining two points in $\mathcal{S}$ crosses $\ell$.) (IMO 2020/6)

**# Return format**
Please return the corresponding difficulty scale (integer) in \box{}

Table 7: Prompt for difficulty estimation using LLM as a judge.

