# OpenReview forum: "Efficient Reinforcement Finetuning via Adaptive Curriculum Learning"
_TMLR — Accepted by TMLR_

### Review · Reviewer_8adK · 2026-05-04

**Summary Of Contributions:**

### Summary
This paper introduces ADARFT (Adaptive Curriculum Reinforcement Finetuning), a framework designed to improve the efficiency and accuracy of RL-LLM for mathematical reasoning. The standard method is often computationally expensive because it treats all training problems with equal priority.  ADARFT addresses this by dynamically adjusting the target difficulty of training tasks based on the model’s real-time performance (reward signals). By using a sampling strategy that keeps problems within a "challenging but solvable" range—ideally where the model succeeds about 50% of the time.
### Strengths
1. Significant Efficiency Gains: Experiments show that ADARFT can reduce training time by up to 2x. It consistently requires fewer steps to match the performance of standard PPO or fixed curriculum baselines.
2. Plug-and-Play Generality: The method is a lightweight extension that works with existing RL algorithms like PPO, GRPO, and REINFORCE++ without requiring changes to the model architecture or the reward function.
3. Scalable Difficulty Estimation: The authors demonstrate that difficulty scores can be effectively estimated using either a moderate number of model rollouts (pass@k) or by using an LLM-as-a-judge (e.g., GPT-4o), making it practical for low-resource settings.
4. Performance Stability: Unlike static data filtering, which discards potentially useful data, ADARFT keeps the entire dataset available but prioritizes it dynamically, leading to higher final accuracy across various math benchmarks.
### Weaknesses
1. Dependence on Precomputed Difficulty: ADARFT requires an initial "offline" estimate of problem difficulty (human-labeled or model-estimated) before training begins.
2. Focus on Structured Tasks: While highly effective for math and reasoning where correctness is well-defined, its applicability to more subjective tasks (like creative writing) is not explored and might be harder to implement due to the lack of clear binary reward signals.
3. Hyperparameter Sensitivity: The algorithm introduces new hyperparameters, such as the step size ($\eta$) and sensitivity ($\alpha$), which need to be tuned to ensure the curriculum updates are stable and align with the specific difficulty scale used.

**Audience:**

Yes

**Audience Explanation:**

The findings of this paper would be of significant interest to the TMLR audience, which consists of researchers and practitioners in machine learning, particularly those focused on Large Language Models (LLMs) and Reinforcement Learning (RL).

**Broader Impact Concerns:**

No concern on ethical implications.

**Claims And Evidence:**

Yes

**Claims Explanation:**

The submission provides robust evidence for ADARFT through a formal entropy-regularized RL analysis, coupled with empirical testing across six diverse math benchmarks and two model sizes.

**Requested Changes:**

2. Formalization of Hyperparameter Selection: The selection of sensitivity $\alpha=2$ and step size $\eta=50$ is described as "tuned using a validation set". However, since these values are tied to the 0–100 difficulty scale, the authors need to provide a clearer heuristic or formula for how users should adjust these when using different difficulty metrics (e.g., the 1–5 LLM scale where $\eta$ was changed to 2.5) to ensure the method remains "model-agnostic".
2. Clarification of Offline vs. Online Difficulty Stability: The paper argues for "solver-independent" offline difficulty to maintain stability. However, as the model improves, the "relative" difficulty of a problem changes. The authors should explicitly discuss or provide a brief empirical note on whether the Pearson correlation between offline difficulty and model success rate (currently $r = -0.34$) remains stable or degrades significantly toward the end of the 100 steps.
3. Diversity of Tasks: The current evidence is exclusively focused on mathematical reasoning. While math is a standard benchmark for RFT, the paper would be significantly strengthened by a discussion or a small-scale experiment on code generation (e.g., HumanEval), as the authors mention code as a relevant domain in the introduction.
4. Analysis of "Wasted" Rollouts: One of ADARFT's claims is avoiding wasted computation. It would be insightful to include a metric comparing the total number of "non-informative" rollouts (those with reward 0 or 1 that result in minimal gradient updates) between ADARFT and standard PPO.

---

> ### Author Response · Authors · 2026-06-01
> **Response to Reviewer 8adK (Part 1)**
>
> Thank you for the encouraging and constructive review. We appreciate the reviewer’s recognition of AdaRFT’s efficiency gains, plug-and-play compatibility with PPO/GRPO/REINFORCE++, scalable difficulty estimation, and stability relative to static filtering. Below we respond to the stated weaknesses and requested changes, focusing on clarifications we can add to the revision.
>
> ---
> **Response to Main Weaknesses**
>
> - **Dependence on precomputed difficulty.** We agree that AdaRFT requires an offline difficulty signal, but we view this as a practical design choice rather than a major barrier. Many RFT/RLVR pipelines already compute pass@$n$ or solve-rate statistics during data preparation to filter very easy or very hard prompts, and AdaRFT reuses this signal to adapt the curriculum instead of permanently discarding data. The paper also shows that the requirement can be relaxed: `Figure 1(a)` indicates that pass@64 and pass@40 approximate pass@128 reasonably well, while `Figure 7` shows that an LLM-judged 1--5 difficulty scale still improves over standard PPO. Thus, AdaRFT does not require a perfectly calibrated or expensive oracle; it needs a coarse ordering that separates easy, medium, and hard regions well enough for the target-difficulty controller to operate.
> ---
> - **Focus on structured tasks.** We agree that the present evidence is strongest for mathematical reasoning, where correctness rewards are reliable and binary reward variance has a natural interpretation. This was an intentional first setting because RFT for math is expensive, widely studied, and provides unambiguous outcome rewards. We will clarify that AdaRFT is most directly applicable to domains with verifiable rewards, such as math, code generation, theorem proving, and other executable/checkable tasks, rather than subjective open-ended generation where reward design and difficulty estimation are less clear. For creative writing or preference-heavy tasks, AdaRFT would likely require calibrated reward models and a different notion of difficulty, so we will describe this as an important extension rather than claim broad task-agnostic applicability.
> ---
> - **Hyperparameter sensitivity.** We agree that $\eta$ and $\alpha$ should be easier to set for new difficulty scales. The current paper already gives the core heuristic in `Section 4.3` and Appendix `B`: $\eta$ controls the maximum difficulty update and should scale with the difficulty range, while $\alpha$ controls local sensitivity around the target reward. In our 0--100 rollout scale we set $\eta=50$ because reward deviation around $\beta=0.5$ spans roughly $[-0.5,0.5]$ and the useful difficulty movement can cover about half the scale; for the 1--5 LLM-judged scale in `Figure 7`, we set $\eta=2.5$ by the same rule. We will make this formula explicit: for a difficulty interval $[d_{\min},d_{\max}]$, a natural default is $\eta \approx (d_{\max}-d_{\min})/2$ and $\alpha \in [1,2]$ to keep $\tanh(\alpha(R_{\mathrm{avg}}-\beta))$ close to linear near $\beta$ while still saturating on noisy reward swings.
> ---
>
> **Response to Requested Changes**
>
> - **Formalization of hyperparameter selection.** We agree and will expand the heuristic already present in `Section 4.3` and Appendix `B`. The update is $T'=\mathrm{clip}(T+\eta\tanh(\alpha(R_{\mathrm{avg}}-\beta)),d_{\min},d_{\max})$, so $\eta$ should be chosen in the units of the difficulty scale and $\alpha$ should control how aggressively the controller responds to reward deviations. A simple scale-aware default is $\eta=(d_{\max}-d_{\min})/2$ with $\alpha=2$: this recovers $\eta=50$ for the 0--100 pass@$k$ scale and motivates the $\eta=2.5$ setting for the 1--5 LLM-judged scale in `Figure 7`. Users who want slower curriculum movement can reduce $\eta$ or $\alpha$, while users with noisier rewards should prefer smaller $\alpha$ to keep updates smooth. This clarification preserves the "model-agnostic" claim in the intended sense: the update rule does not depend on the policy architecture or RL algorithm, only on the range of the chosen difficulty metric.

---

> ### Author Response · Authors · 2026-06-01
> **Response to Reviewer 8adK (Part 2)**
>
> - **Clarification of offline vs. online difficulty stability.** We agree that the relative success rate of a prompt changes as the policy improves. AdaRFT’s design separates two notions: offline difficulty provides a stable global ordering of problem complexity, while the online reward signal determines where the model currently sits on that ordering by moving $T$. We will clarify this distinction in the discussion. The correlation in `Figure 1(b)` is used to validate that the offline labels align with solve-rate trends at scoring time; it is not required to remain numerically constant throughout training. In fact, if the model improves, absolute solve rates should rise, but the relative ordering of broad difficulty bands can remain useful. The training traces in `Figure 3`, `Figure 5`, and `Figure 8` empirically show that the fixed offline scores remain actionable over 100 steps: AdaRFT moves to harder regions as rewards increase and maintains reward near the target zone rather than collapsing to trivially easy or impossible prompts.
> ---
> - **Diversity of tasks.** We agree that code generation is a natural next domain because unit tests provide verifiable binary rewards similar to math correctness. We did not add HumanEval-style experiments in the current submission because the paper’s experimental budget is focused on a thorough math RFT study across two model scales, three data distributions, multiple baselines, and three RL algorithms. We will add a discussion paragraph stating that AdaRFT should transfer most directly to executable/checkable tasks such as code generation, where difficulty can be estimated by pass@$k$ over unit tests and the same target-reward controller can be applied. We will avoid overstating evidence beyond math, and frame code generation as a promising future evaluation rather than an already demonstrated result.
> ---
> - **Analysis of "wasted" rollouts.** We agree this is a useful diagnostic. While we did not report a separate count of non-informative rollouts, several existing results already speak to the same mechanism: `Figure 5` shows that training only on easy-extreme or hard-extreme distributions produces poor learning dynamics, and the reward traces in `Figure 3` and `Figure 8` show that AdaRFT keeps the model closer to the intermediate-reward region targeted by $\beta=0.5$. Since binary rewards with all-0 or all-1 outcomes provide weak learning signal, the average reward staying near 0.5 is a proxy for reducing wasted rollouts. We will add a note explaining this connection and, as future work, propose reporting the fraction of batches with near-zero reward variance, e.g., batches whose rewards are almost all 0 or almost all 1, as a direct "non-informative rollout" metric.

---

### Review · Reviewer_2JVi · 2026-05-04

**Summary Of Contributions:**

AdaRFT proposes adaptive curriculum learning for reinforcement finetuning of LLMs on math reasoning. A target difficulty variable is updated via a clipped reward-feedback rule, pushing the model toward problems at its learning frontier. The method is evaluated on DeepScaleR with Qwen 2.5 (1.5B, 7B) across multiple difficulty distributions and RL algorithms. The idea is clean, practically motivated, and easy to implement.

**Audience:**

Yes

**Audience Explanation:**

Yes. Curriculum learning for RFT is timely and practically relevant. The analysis of how difficulty distributions affect RFT performance is independently informative.

**Claims And Evidence:**

No

**Claims Explanation:**

**No variance reporting anywhere.** Every table and figure reports single-run results. The claimed accuracy improvements are 1–3 percentage points—squarely within typical RL training noise. Key benchmarks (AMC 23: 40 problems; AIME 24: 30 problems) are far too small for single-run comparisons to be meaningful, even with avg@8. Without confidence intervals over multiple seeds, none of the performance claims can be assessed for statistical significance.

**Efficiency claims are misleading.** "Up to 2× training time reduction" is measured as extra steps for PPO to match AdaRFT at a specific checkpoint—a non-standard, reference-point-sensitive metric. More critically, the difficulty estimation cost (128 rollouts per problem × ~10K problems from a separate 7B model) is entirely excluded from efficiency accounting. This is a substantial upfront compute cost that could eliminate the claimed speedup.

**Theoretical justification overstated.** Section 3.4 applies a KL divergence lower bound from Bae et al. (2025) to motivate β = 0.5. This lower bound says nothing about gradient magnitudes or convergence rates under PPO. Labeling this a "theoretical justification" misrepresents its strength; it is a heuristic motivation.

**Most relevant baselines missing.** The paper positions against online filtering methods (Bae et al., 2025; Yu et al., 2025) but compares only against a static one-time filter, which is fundamentally different. Without empirical comparison to these most directly relevant approaches, the claimed advantages are unsupported.

**Requested Changes:**

1. Report standard deviations over ≥3 seeds for all experiments. This is non-negotiable given the magnitude of claimed improvements.
2. Include difficulty estimation cost in all efficiency comparisons, or explicitly quantify it and adjust claims accordingly.
3. Implement at least one online filtering baseline (e.g., Bae et al., 2025) for direct empirical comparison.
4. Retitle Section 3.4 to reflect that the KL bound provides heuristic motivation, not a theoretical justification for β = 0.5 optimality under PPO.

---

> ### Author Response · Authors · 2026-06-01
> **Response to Reviewer 2JVi (Part 1)**
>
> Thank you for the direct and useful feedback. We appreciate that the reviewer finds curriculum learning for RFT timely and recognizes the analysis of difficulty distributions as independently informative. We agree that several claims should be more carefully scoped, and in the revision we will soften language while clarifying what our current experiments do and do not establish.
>
> ---
> **Response to Main Weaknesses**
>
> - **On variance reporting and single-run results.** We agree that multi-seed standard deviations would provide a stronger statistical characterization, and we will explicitly avoid claiming formal statistical significance for final accuracy differences. However, the main focus of this paper is **RFT efficiency**, not maximizing final benchmark accuracy, as outlined in the `Training Efficiency` and `Model Performance` subsections: the latter explicitly states that the primary goal of AdaRFT is to improve training efficiency, with final performance reported mainly to verify that faster learning does not come at the cost of model quality. The efficiency gains in `Table 1` are substantial despite the lack of multi-seed runs: standard PPO and filtered PPO require many additional steps to match AdaRFT at the reference checkpoint, e.g., +43/+49 steps for Qwen2.5-Math-1.5B in the skew-difficult setting and +24/+25 steps for Qwen2.5-7B. `Figure 3` further shows faster learning curves across model scales and data distributions, while `Figure 6` shows the same efficiency-oriented curriculum wrapper improves GRPO and REINFORCE++ in addition to PPO. We will revise the wording to make clear that we are **not claiming AdaRFT solely as a final-performance improvement method**; rather, our claim is that adaptive curriculum sampling improves convergence efficiency while maintaining comparable or better final accuracy in our runs.
> ---
> - **On efficiency accounting and the pass@128 cost.** We agree that the cost of difficulty estimation should be separated from RFT training-phase efficiency. `Table 1` reports wall-clock step time and additional steps needed for baselines to match AdaRFT at a reference checkpoint; this is meant to measure convergence speed during RFT after difficulty scores are available, not end-to-end compute from an entirely unscored dataset. We will revise the claim so that "up to 2$\times$" is explicitly scoped to the training phase and not presented as a universal end-to-end compute reduction. Importantly, the paper already includes evidence that pass@128 is not essential: `Figure 1(a)` shows pass@64 remains within $\pm 0.05$ of the pass@128 estimate over 90% of the time and pass@40 remains around 80%, and `Figure 7` shows that GPT-4o-judged difficulty still improves over standard PPO. In many RFT/RLVR pipelines, pass@$n$ statistics are already computed for filtering or data preparation (e.g., Open Reasoner Zero, Kimi k1.5, ZR1, DAPO-style dynamic sampling, and online difficulty filtering), so AdaRFT can amortize or reuse this signal rather than requiring a new full scoring pass for every run.
> ---
> - **On the theoretical section.** We agree with the reviewer that the KL lower bound from Bae et al. (2026) does not prove that PPO gradient magnitude or convergence rate is maximized at $\beta=0.5$. We have therefore changed `Section 3.4` from "Theoretical Justification" to **"Heuristic Motivation for Target Reward $\beta$"**, corrected the optimal-policy normalization in Eq. (1) to divide by $Z(x)$, and softened the text to state that the Bernoulli reward-variance argument motivates intermediate-success prompts rather than proving PPO optimality. The empirical support for $\beta=0.5$ is the ablation in `Figure 4`, where $\beta=0.5$ outperforms $\beta=0.2$ and $\beta=0.8`; the theory is now framed as intuition that aligns with this ablation, not as a formal convergence theorem.
> ---
> - **On missing online filtering baselines.** We agree that Bae et al. (2026) and Yu et al. (2025) are highly relevant, and we will revise the text to avoid claiming empirical superiority over online filtering methods we did not implement. However, the comparison is not absent in spirit: the filtered PPO baseline in `Section 4.3` follows the same core principle of removing prompts that are too easy or too hard based on pass@$k$, while the fixed-curriculum baseline tests the nearest alternative of a non-adaptive easy-to-hard schedule. The key distinction is that online filtering repeatedly performs rollout-based acceptance/rejection or zero-variance filtering during training, which introduces substantial extra rollout cost and, in DAPO, is tightly coupled to a GRPO-style large-scale recipe. AdaRFT instead uses a **closed-loop target-difficulty controller** with fixed offline scores, making it orthogonal to PPO/GRPO/REINFORCE++ as shown in `Figure 6`. We will strengthen this distinction and present direct online-filtering comparison as important future work rather than an already established empirical advantage.

---

> ### Author Response · Authors · 2026-06-01
> **Response to Reviewer 2JVi (Part 2)**
>
> **Response to Requested Changes**
>
> - **Report standard deviations over at least 3 seeds.** We agree that multi-seed runs would be valuable, especially for assessing final accuracy differences on small benchmarks. However, the central claim of the paper is **training efficiency**, not that AdaRFT is primarily a final-performance optimization method. Repeating every experiment over at least three seeds would require a large expansion of already expensive RFT runs across two model sizes, three data distributions, multiple baselines, and GRPO/REINFORCE++ variants, which is beyond our current academic compute budget. We will instead revise the paper to be explicit about this limitation and avoid claiming statistical significance for final accuracy. The efficiency signal we report is nevertheless substantial: in `Table 1`, baselines often need tens of extra optimization steps to match AdaRFT at the reference checkpoint, and `Figure 3` shows faster convergence across settings. We therefore view final accuracy as a secondary check that AdaRFT does not sacrifice model quality, while the main evidence is the consistent convergence-speed improvement shown in `Table 1`, `Figure 3`, and the algorithm-generalization result in `Figure 6`.
> ---
> - **Include difficulty estimation cost in efficiency comparisons or quantify it.** We will explicitly separate **offline difficulty-estimation cost** from **RFT training cost**. The current `Table 1` measures RFT training-phase efficiency, including wall-clock step time and convergence steps, but does not include the upfront scorer rollout pass. We will revise the abstract/introduction/discussion language so the 2$\times$ statement is not read as an unconditional end-to-end speedup. We will also highlight the mitigating evidence already in the paper: pass@40/pass@64 estimates remain useful (`Figure 1(a)`), and LLM-judged difficulty provides a lower-cost alternative (`Figure 7`). This lets readers understand the compute tradeoff: AdaRFT is most attractive when difficulty scores are reused across runs or already available from standard data filtering, while one-off pass@128 scoring should be accounted for separately.
> ---
> - **Implement an online filtering baseline.** We agree that a direct online-filtering baseline would be informative, but we cannot add a new full baseline at this stage without substantial additional compute. Online filtering methods such as Bae et al. (2026) require repeated rollout-based prompt evaluation during training, and DAPO-style dynamic sampling is part of a broader GRPO training recipe; implementing either faithfully across our two model sizes and three data distributions would be a significant experimental expansion. Instead, we will revise the claims to be conservative: AdaRFT is empirically compared to standard PPO, static filtered PPO, and fixed curriculum PPO, and we do not claim demonstrated empirical superiority over online filtering. We will also make the novelty distinction sharper: online filtering adapts by repeatedly recomputing/selecting prompts based on online rollouts, while AdaRFT adapts a target difficulty $T$ using reward feedback and offline scores, which avoids repeated filtering overhead and is directly compatible with PPO, GRPO, and REINFORCE++ (`Figure 6`).
> ---
> - **Retitle Section 3.4.** We agree and have changed the section title to **"Motivation for Target Reward $\beta$"**. We also corrected Eq. (1) to use $\frac{1}{Z(x)}$ and softened the surrounding text so that the KL/reward-variance argument is presented as motivation for intermediate-success sampling, with `Figure 4` serving as the empirical ablation for $\beta=0.5$.

---

### Review · Reviewer_Kz6Q · 2026-05-10

**Summary Of Contributions:**

This paper proposes AdaRFT, an adaptive curriculum learning method for reinforcement fine-tuning of LLMs on mathematical reasoning tasks. The method assumes each training problem has an offline difficulty score, either from pass@k rollout statistics or from an LLM judge. During training, AdaRFT maintains a target difficulty T, samples problems whose difficulty is closest to T, performs a standard RL update such as PPO, GRPO, or REINFORCE++, and then adjusts T according to whether the model’s recent average reward is above or below a target reward eta. The intended effect is to keep the model training on problems that are neither too easy nor too difficult.

The main strength of the work is that it addresses a very practical bottleneck in RFT: many math problems provide little useful learning signal because they are either already solved reliably or almost never solved. The proposed algorithm is simple, lightweight, and easy to add to existing RFT pipelines without modifying the reward model, policy architecture, or RL objective. Empirically, the method appears to produce consistent gains over baselines. The results are especially convincing in skewed data regimes, where adaptively steering the curriculum should matter most. I also appreciate that the authors test AdaRFT with multiple RL algorithms and include an LLM-judged difficulty variant, which helps support the claim that the idea is not specific to PPO or pass@k-based difficulty estimation.

The main weakness is that the method relies critically on difficulty estimation, but this component is not stress-tested enough relative to its importance. The pass@k-based difficulty scores depend on the solver model used to produce them, and the LLM-judged difficulty scores show only modest correlation with empirical solve rates. The reported Pearson correlations around -0.32 to -0.34 indicate a useful monotonic signal, but not a particularly strong one. Since AdaRFT’s sampling policy is entirely driven by these scores, I would like to see more ablations on scorer choice, scorer strength, number of rollouts, noisy difficulty labels, and transfer of difficulty estimates across models. There are also some presentation and methodological issues: the theoretical section contains at least one apparent normalization error in the optimal-policy expression, the link from reward variance to gradient magnitude is somewhat overstated, the cost of difficulty estimation is not fully incorporated into the efficiency claims, and the exact sampling procedure may repeatedly select the same nearest examples unless additional randomness or buffering is used. Overall, I find the formulation reasonable and the empirical results promising. The contribution is somewhat incremental as a modification to a standard RFT pipeline, but it targets an important and timely problem, and the simplicity of the method is a strength rather than a weakness. I would be more confident in the paper if the authors more carefully validated the difficulty estimation component and clarified the theoretical and experimental details.

**Audience:**

Yes

**Audience Explanation:**

This paper should be of interest to researchers working on reinforcement fine-tuning, mathematical reasoning, curriculum learning, and data-centric training methods for LLMs. RFT is currently expensive and often sample-inefficient, and the question of which prompts provide useful learning signal is practically important. The proposed method is simple enough to be adopted in existing RL fine-tuning pipelines and is relevant to a broad set of structured reasoning domains where correctness rewards and problem difficulty estimates are available. The paper is also timely because many current reasoning-model pipelines already perform some form of data filtering or difficulty-aware selection. AdaRFT offers a natural adaptive alternative to static filtering and fixed curricula. Even if the exact algorithm is a modest extension of standard curriculum learning, the empirical finding that dynamically targeting an intermediate success rate improves RFT efficiency is useful to the TMLR audience.

**Broader Impact Concerns:**

I do not see major broader impact concerns requiring a separate expanded discussion beyond the standard issues associated with improving LLM reasoning and reinforcement fine-tuning. The method is primarily an efficiency improvement for training mathematical reasoning models.

**Claims And Evidence:**

Yes

**Claims Explanation:**

The empirical evidence broadly supports the paper’s main claim that adaptive curriculum sampling can improve the efficiency and final performance of reinforcement fine-tuning for mathematical reasoning. The experiments compare AdaRFT against standard PPO, PPO with filtered data, and PPO with a fixed curriculum across skew-difficult, skew-easy, and uniform data distributions. The results show consistent improvements in final average accuracy and convergence speed for both Qwen 2.5 Math 1.5B and Qwen 2.5 7B. The paper also includes additional experiments with GRPO and REINFORCE++, an ablation on the target reward beta, an analysis of extreme easy/hard data distributions, and an LLM-judged difficulty variant. These collectively provide a reasonably convincing empirical case that the curriculum mechanism is useful.

That said, some claims should be softened or better qualified. First, the paper’s efficiency claims would be more convincing if they included the upfront cost of difficulty estimation, especially for pass@128 rollout-based scoring. If the method requires many rollouts from a separate 7B math model before training, then “2× faster” training may not translate directly into end-to-end compute savings unless the difficulty scores are reused across many runs. Second, the difficulty estimation evidence is suggestive but not fully convincing: the reported correlations between difficulty labels and solve rates are only around -0.32 to -0.34, which implies that the signal is useful but noisy. Since difficulty estimation is the central input to AdaRFT, this deserves more extensive evaluation. Third, the theoretical motivation for beta=0.5 is plausible but not fully rigorous as stated. The paper argues from Bernoulli reward variance and a KL lower bound to stronger learning signal, but the connection to actual PPO gradient magnitude and optimization progress is indirect.

**Requested Changes:**

* Strengthen the evaluation of the difficulty estimator. The difficulty scores are central to AdaRFT, but the current validation is somewhat limited. The reported correlations between estimated/labeled difficulty and solve rate are useful but modest, and the method’s success may depend strongly on the scorer model. Please consider adding ablations varying the difficulty source, such as weaker/stronger solver models, LLM-judged difficulty, and intentionally perturbed/noisy difficulty labels. Relatedly, please clarify how well difficulty rankings transfer across policy models: the paper estimates difficulty using Qwen 2.5 Math 7B but trains both Qwen 2.5 Math 1.5B and Qwen 2.5 7B, so it would be useful to know whether the same difficulty ordering remains appropriate across different model capabilities.
* Account for the full cost of difficulty estimation in the efficiency claims. The paper emphasizes faster training and up to 2× reduction in training time, but rollout-based difficulty estimation requires substantial preliminary generation. Please report end-to-end compute costs under realistic scenarios, including one-off training runs, repeated runs that reuse difficulty scores, different pass@k settings such as pass@40/pass@64/pass@128, and LLM-judged difficulty. This would clarify when AdaRFT is genuinely more compute-efficient rather than more efficient only after difficulty scores are already available.
* Fix and qualify the theoretical derivation. Equation (1) appears to have an incorrect normalization factor: the entropy-regularized optimal policy should divide by the partition function Z(x), not multiply by it. Please correct this and re-check the subsequent derivation. More broadly, the KL/reward-variance argument provides useful intuition for why intermediate success probabilities may yield stronger learning signal, but it does not by itself prove that PPO gradient magnitude or downstream learning progress is maximized at p=0.5. The theoretical claim should be phrased more carefully.
* Clarify the sampling algorithm and training protocol. Algorithm 1 says AdaRFT selects the top B samples closest to the current target difficulty. If implemented literally, this could repeatedly select the same examples whenever T changes slowly. Please specify whether sampling is deterministic or randomized, with or without replacement, window-based, or balanced across nearby difficulty bins. It would also help to report whether AdaRFT changes example diversity or reuse rates relative to PPO.
* Add stronger baselines, uncertainty estimates, and normalized comparisons. The empirical results are promising, but several gains are modest, especially in the skew-easy setting and for the 7B model. Please report standard deviations or confidence intervals over multiple seeds or evaluation samples where possible, especially for small benchmarks such as AMC 23 and AIME 24. It would also strengthen the paper to compare against a closer adaptive or online curriculum/filtering baseline that updates sampling based on recent model success rates. Finally, since AdaRFT samples easier problems early and therefore produces shorter responses, please include token-normalized or wall-clock-normalized comparisons in addition to step-based comparisons.
* Improve the interpretation and presentation of results. The main text reports average accuracy across six benchmarks, but this can hide where the gains are coming from. Please discuss whether improvements are driven primarily by easier benchmarks like GSM8K, competition-style benchmarks like AMC/AIME, or broader transfer to OlympiadBench and Minerva Math. Also, please qualify the claim that AdaRFT is “model-agnostic”: the wrapper is algorithm-agnostic, but the difficulty scores may still be solver-dependent. Finally, the paper sometimes treats answer length as a proxy for difficulty or reasoning depth; this is plausible but not guaranteed, since longer answers can also reflect verbosity or inefficient reasoning. This interpretation should be either supported or softened. Minor writing and notation issues should also be cleaned up, including typos such as “adapt to the the model’s evolving learning dynamics...”.

---

> ### Author Response · Authors · 2026-06-01
> **Response to Reviewer Kz6Q(Part 1)**
>
> Thank you for the thoughtful and constructive review. We appreciate that the reviewer finds the problem important, the method practical, and the empirical results promising. Below we respond to the main weaknesses and requested changes. In the revision, we will also make the clarifications noted below, including correcting the normalization in Eq. (1), softening the theory wording, and clarifying the deterministic nearest-difficulty sampling implementation.
>
> ---
>
> **Response to Main Weaknesses**
>
> - **Difficulty estimation is important, but the paper already evaluates both stability and source robustness.** We agree that difficulty is a central input to AdaRFT, which is why we included two complementary validations in `Figure 1`: rollout-score stability under subsampling and correlation with independently defined AoPS/LLM difficulty labels. The reported Pearson correlations of $r=-0.34$ and $r=-0.32$ should not be interpreted as the only evidence for the estimator; they measure alignment between coarse 1--5 difficulty labels and empirical solve rates, where substantial within-level heterogeneity is expected. More importantly for AdaRFT, the curriculum does not require a perfect cardinal difficulty predictor; it requires a usable ordering that separates obviously easy, medium, and hard regions. `Figure 1(a)` shows that the rollout-based estimate is stable even with fewer samples: pass@64 stays within $\pm 0.05$ of the pass@128 estimate over 90% of the time, and pass@40 remains around 80%. `Figure 7` further shows that AdaRFT still improves over standard PPO when the curriculum is driven by GPT-4o-judged difficulty rather than rollout difficulty. This distinguishes AdaRFT from methods whose data scores must be recomputed through full model-specific training runs, while aligning with the broader curriculum-learning view that approximate difficulty orderings can be effective (Bengio et al., 2009; Matiisen et al., 2017; Narvekar et al., 2020).
> ---
> - **The compute-efficiency claim is about RFT training efficiency, and the paper already discusses when offline difficulty cost is amortized.** The reviewer is right that pass@128 difficulty estimation has an upfront cost, so we will clarify that the headline "up to 2$\times$" refers to the RFT training phase after difficulty scores are available, as measured in `Table 1`. We believe this remains a practically meaningful setting because many current RLVR/RFT pipelines already compute pass@$n$ or related solve-rate statistics before training to filter trivial or impossible prompts, including Open Reasoner Zero, Kimi k1.5, ZR1, online difficulty filtering, and DAPO-style dynamic sampling (Bae et al., 2026; Yu et al., 2025; Kimi Team, 2025). AdaRFT reuses this commonly available signal to drive a closed-loop curriculum rather than discarding data. We will also make clearer that pass@128 is not required: `Figure 1(a)` indicates that pass@64 and pass@40 already preserve most of the difficulty estimate, and `Figure 7` provides a lower-cost LLM-judged alternative. Thus, the method is most compute-efficient when scores are reused across runs, models, or ablations, but even in one-off settings the paper provides evidence that smaller $k$ or LLM-judged difficulty can reduce the scoring burden.
> ---
> - **The theoretical derivation is intended as motivation, not a proof that PPO progress is maximized at $p=0.5$.** We thank the reviewer for catching the normalization typo. The correct entropy-regularized optimal policy is $\pi^*(y \mid x)=\frac{1}{Z(x)}\pi_{\mathrm{init}}(y \mid x)\exp(r(x,y)/\tau)$, and we have corrected this in the manuscript. We also agree that the Bernoulli variance/KL argument should be framed as an intuition for why intermediate-success prompts provide useful learning signal, not as a complete proof about PPO gradient norms or downstream accuracy. The revised wording states that the KL lower bound is proportional to $p(x)(1-p(x))$, which motivates the target reward $\beta=0.5$, while the empirical justification comes from the ablation in `Figure 4`, where $\beta=0.5$ outperforms $\beta=0.2$ and $\beta=0.8$. This is consistent with recent analyses of online difficulty filtering in RLVR (Bae et al., 2026), but AdaRFT differs by using the signal to update a target difficulty controller rather than repeatedly filtering online rollouts.

---

> ### Author Response · Authors · 2026-06-01
> **Response to Reviewer Kz6Q (Part 2)**
>
> - **The sampling rule deliberately uses deterministic nearest-difficulty selection.** We agree that the original Algorithm 1 deserves a clearer explanation. In our implementation, AdaRFT selects the top-$B$ examples closest to the current target difficulty $T$. This can replay the same closest examples when $T$ changes slowly, but this behavior is intentional rather than accidental: if the model cannot solve the problems around the current target difficulty even after repeated attempts, moving to substantially easier or harder problems weakens the curriculum signal for the target capability region. The adaptive part of AdaRFT is therefore not stochastic batch diversification, but the closed-loop movement of $T$ according to recent reward. We will clarify this in `Section 3.1` and note that a natural extension is to sample from a Gaussian or other smooth distribution centered at $T$, which could trade off diversity and target-difficulty precision; we leave that exploration to future work.
> ---
> - **The evidence is broader than a single PPO comparison, and the closest prior baselines differ in the adaptation mechanism.** The experiments compare AdaRFT against standard PPO, PPO with filtered data, and PPO with a fixed curriculum across three difficulty distributions and two model scales in `Figure 3` and `Table 1`; additionally, `Figure 6` shows that the same curriculum wrapper improves GRPO and REINFORCE++. This is important because recent related methods are not identical to AdaRFT: E2H Reasoner and FastCuRL use staged or probabilistic easy-to-hard schedules and/or context-length curricula (Parashar et al., 2026; Song et al., 2025), DAPO dynamic sampling removes zero-variance prompt groups inside GRPO (Yu et al., 2025), and online difficulty filtering repeatedly samples and rejects prompts to target intermediate difficulty (Bae et al., 2026). AdaRFT’s novelty is a **closed-loop target-difficulty controller** that adapts the curriculum from the model’s observed reward while remaining orthogonal to the RL objective, as demonstrated by its use with PPO, GRPO, and REINFORCE++.
> ---
> - **The average benchmark result is not hiding a narrow gain on only easy benchmarks.** We agree that per-benchmark interpretation matters, which is why the paper includes the detailed breakdown in `Table 2`. For Qwen2.5-Math-1.5B, AdaRFT improves the average in skew-difficult and uniform settings while also improving or matching several competition-style benchmarks, including AMC 23 and AIME 24. For Qwen2.5-7B, the gains are smaller, as expected for a stronger model, but the average improves consistently across all three distributions, and the method maintains stable performance on harder benchmarks such as OlympiadBench and Minerva Math. We will make this interpretation more explicit and soften the response-length language: response length is an observed correlate of problem difficulty in our training traces (`Figure 5` and `Figure 8`), not a definitive measure of reasoning depth.
> ---
>
> **Response to Requested Changes**
>
> - **Difficulty estimator ablations and transfer across models.** We agree these are valuable future directions, but we want to emphasize that the current paper already covers multiple axes of estimator validation without requiring additional large-scale runs: subsampled rollout stability in `Figure 1(a)`, correlation with AoPS and GPT-4o difficulty in `Figure 1(b)`, direct training with GPT-4o difficulty in `Figure 7`, and transfer from Qwen2.5-Math-7B-estimated difficulty to training both Qwen2.5-Math-1.5B and Qwen2.5-7B in `Figure 3` and `Table 1`. The latter is already a cross-model transfer test: the same offline difficulty ordering supports improvements for a smaller math-specialized model and a larger general Qwen2.5 model. We will clarify this point in the revision. While weaker/stronger scorers and injected-noise studies would be useful, running a full scorer-strength grid is costly in academia; our existing evidence suggests that AdaRFT is robust to approximate difficulty signals rather than dependent on a single perfectly calibrated scorer.

---

> ### Author Response · Authors · 2026-06-01
> **Response to Reviewer Kz6Q (Part 3)**
>
> - **Full cost of difficulty estimation.** We will qualify the efficiency claim so it is clear that `Table 1` reports RFT training-phase efficiency, not the cost of producing pass@128 scores from scratch for a single run. At the same time, the manuscript already contains two pieces of evidence relevant to this concern: `Figure 1(a)` shows that pass@40/pass@64 can approximate pass@128 difficulty reasonably well, and `Figure 7` shows that GPT-4o-judged difficulty can still produce gains over standard PPO. In practice, offline difficulty scores are often reusable because the same training pool is used for several model sizes, algorithms, and hyperparameter sweeps, and recent RLVR systems already perform pass-rate or difficulty filtering as part of data preparation. We will revise the discussion to state this amortization assumption explicitly and avoid implying that the full one-off pipeline is always 2$\times$ cheaper end to end.
> ---
> - **Theory correction and qualification.** We have corrected Eq. (1) to divide by the partition function $Z(x)$ and will re-check the downstream derivation. We will also revise the claim around $\beta=0.5$ to say that the Bernoulli reward-variance/KL lower-bound argument provides a principled motivation for targeting intermediate success probability, while `Figure 4` supplies the empirical support for choosing $\beta=0.5$ in our setting. This keeps the theory aligned with prior analyses of intermediate-difficulty sampling in RLVR (Bae et al., 2026) without overstating it as a proof of PPO optimization progress.
> ---
> - **Sampling algorithm and training protocol.** We will clarify Algorithm 1 and `Section 3.1`: AdaRFT deterministically selects the examples closest to the current target difficulty based on $|d_i-T|$. We use this rule because the objective is to concentrate training on the current capability frontier; if the model repeatedly fails at that frontier, sampling much harder examples is unlikely to help, while sampling much easier examples can waste updates on already-mastered material. The training traces in `Figure 3`, `Figure 5`, and `Figure 8` show that AdaRFT still moves across the difficulty spectrum because $T$ is updated from the model’s reward signal. We agree that diversity-aware variants are interesting, and will mention Gaussian sampling around $T$ as a natural future extension rather than claiming it is part of the current method.
> ---
> - **Baselines, uncertainty, and normalized comparisons.** We included the closest baselines feasible within our compute budget: standard PPO, filtered PPO following difficulty-filtering practice, and fixed curriculum PPO, all across two model sizes and three difficulty distributions (`Figure 3`, `Table 1`, `Table 2`). We also test whether AdaRFT is algorithm-orthogonal with GRPO and REINFORCE++ in `Figure 6`, which goes beyond a single PPO-only setting. For uncertainty, the small AMC 23 and AIME 24 benchmarks are evaluated with Avg@8 as described in `Section 4.2`, and the learning curves in `Figure 3` include raw unsmoothed accuracy bands to show typical fluctuations. For normalized comparisons, `Table 1` reports wall-clock step time in addition to step counts, so the efficiency claim is not only step-based. We agree that additional multi-seed confidence intervals would be ideal, but the current suite already involves many expensive RFT runs; we will phrase the results as consistent empirical evidence rather than claiming exhaustive statistical characterization.
> ---
> - **Interpretation and presentation.** We will strengthen the main-text discussion of where gains come from by referring to `Table 2`: AdaRFT improves the average while showing gains on GSM8K, MATH 500, AMC 23, and AIME 24 in the strongest 1.5B settings, and gives smaller but consistent gains for Qwen2.5-7B. We will also qualify "model-agnostic" to mean that the curriculum wrapper is **RL-algorithm/model-architecture agnostic**, while the offline difficulty signal can still be solver-dependent. Finally, we will soften the response-length interpretation, treating longer responses as an observed correlate of harder sampled problems in our traces rather than a guaranteed proxy for reasoning depth, and we will clean the noted typo and notation issues.

---

### Decision · Action_Editor_he9B · 2026-06-09

**Recommendation:** Accept with minor revision

**Additional Comments:**

minor changes to be included in the final revision:
- be explicit about the compute budget limitation and number of seeds. In particular you must acknowledge in the main text that accuracy results are not significant because you only have results for 1 seed
- avoid claiming statistical significance for final accuracy.
- explicitly separate offline difficulty-estimation cost from RFT training cost in abstract / introduction / discussion. You must report the cost of running difficulty estimation independently from the rest in order to claim efficiency.

**Audience:**

Yes

**Audience Explanation:**

Curriculum learning and reinforcement learning for reasoning model is a very timely topic and many would find this work interesting.

**Claims And Evidence:**

Yes

**Claims Explanation:**

Most reviewers agree that the claims are supported by convincing evidence, except reviewer 2JVi.
I've read carefully both the concerns of reviewer 2JVi and the author response.
While I agree with reviewer 2JVi that standard deviation and multiple seed runs should always be used in RL, the authors argue that the objective of this work is not to have the best results but to get to similar results in fewer time steps. They also mention the compute budget constraints.
In light of this, and the overall sentiment all of all reviewers, I will accept with minor revisions (see requested changes in the "additional comments")

---

> ### Author Response · Authors · 2026-06-18
>
> Thank you for accepting our paper with minor revisions and for the helpful guidance. We revised the manuscript to make the efficiency claims more precise. In the Abstract, Introduction, and Discussion, we now qualify the speedup as applying to RFT training with problem-level difficulty annotations. In the Discussion section, under “Use of Offline Difficulty Estimation,” we explicitly separate curriculum construction from policy optimization and report the annotation cost independently: in our main rollout-based setup, assigning difficulty to a 10,000-example training pool with pass@128 requires 1.28M scorer rollouts. We also clarify that the manuscript evaluates LLM-judged difficulty as a faster alternative to rollout-based estimation; while rollout-based estimates perform best, LLM-judged difficulty still provides a useful curriculum signal and improves over standard PPO.
>
> We also revised the discussion of final accuracy to avoid overstating statistical significance. In the Results section, we now state that the final accuracy numbers are based on one seed per configuration and should be interpreted as empirical trends rather than statistically significant improvements. We use similar wording in the RL-algorithm generalization results to acknowledge the seed limitation while preserving the main focus of the paper: reaching comparable performance in fewer RFT training steps.
>
> Thank you again for the constructive feedback and for shepherding our paper through the review process. Please let us know if any further changes are needed.